# Actomyosin organelle functions of SPIRE actin nucleators precede animal evolution
Martin Kollmar [1,11] ✉, Tobias Welz[2,11], Aishwarya Ravi[3], Thomas Kaufmann [4], Noura Alzahofi[5,7], Klas Hatje [1,8], Asmahan Alghamdi [5,9], Jiyu Kim[2,10], Deborah A. Briggs[5], Annette Samol-Wolf[2], Olena Pylypenko [6], Alistair N. Hume [5], Pawel Burkhardt[3], Jan Faix[4] & Eugen Kerkhoff [2] ✉

An important question in cell biology is how cytoskeletal proteins evolved and drove the development of novel structures and functions. Here we address the origin of SPIRE actin nucleators. Mammalian SPIREs work with RAB GTPases, formin (FMN)-subgroup actin assembly proteins and class-5 myosin (MYO5) motors to transport organelles along actin filaments towards the cell membrane. However, the origin and extent of functional conservation of SPIRE among species is unknown. Our sequence searches show that SPIRE exist throughout holozoans (animals and their closest single-celled relatives), but not other eukaryotes. SPIRE from unicellular holozoans (choanoflagellate), interacts with RAB, FMN and MYO5 proteins, nucleates actin filaments and complements mammalian SPIRE function in organelle transport. Meanwhile SPIRE and MYO5 proteins colocalise to organelles in *Salpingoeca rosetta* choanoflagellates. Based on these observations we propose that SPIRE originated in unicellular ancestors of animals providing an actin-myosin driven exocytic transport mechanism that may have contributed to the evolution of complex multicellular animals.

The directed intracellular transport of proteins, RNAs and small molecules towards the cell membrane forms the basis of the structural diversity, polarisation and communication of eukaryotic cells. The crucial step of transport and directed localisation of these factors is facilitated by cytoskeletal components. In eukaryotic cells, proteins destined for integration into the plasma membrane and secretion into the extracellular space are sorted at the trans-Golgi network into membrane-bound vesicles and subsequently transported by motor proteins along microtubule and actin filament tracks towards the cell periphery[1].

In animal cells the model of highways and local roads describes the cooperative transport of cargo along microtubule and actin tracks[2–5]. This model suggests that in animal cells microtubules enable long and fast transport (highways) whereas the dynamic actin cytoskeleton provides slower, but very flexible local delivery of cargo (local roads). In plants and fungi, actin filaments mediate both, fast long-range and slow short-range transport processes[6,7]. Class-11 and class-5 myosin motor proteins

(MYO11, MYO5) are the major actin-based transporters in plants (MYO11), fungi (yeast MYO2p, MYO4p), and animal cells (MYO5, MYO5A, 5B, 5 C), respectively[5,8].

Studies in mammalian oocytes and melanocytes have unveiled an actin-myosin (actomyosin)-driven long-range transport mechanism in animal cells, that may operate alongside the better characterised transport of organelles along cortical actin cytoskeleton tracks[9–11]. For this mode of transport actin tracks are generated directly at organelle membranes and class-5 myosin (MYO5) motor proteins, also attached to the organelle, may then drive the motility along the vesicle-originated actin meshwork. This transport is directed towards the plasma membrane and was designated a centrifugal dispersion mechanism[11]. In mouse skin melanocytes, centrifugal dispersion transports RAB27A (RAB: RAS-related in brain) positive melanosomes to the cell periphery prior to their intercellular transfer to keratinocytes where they accumulate in the perinuclear cytoplasm and provide UV light protection[4,11].

[1]Group Systems Biology of Motor Proteins, Department of NMR-based Structural Biology, Max-Planck-Institute for Biophysical Chemistry, Göttingen, Germany. [2]Molecular Cell Biology Laboratory, Department of Neurology, University Hospital Regensburg, Regensburg, Germany. [3]Michael Sars Centre, University of Bergen, Bergen, Norway. [4]Institute for Biophysical Chemistry, Hannover Medical School, Hannover, Germany. [5]School of Life Sciences, University of Nottingham, Nottingham, UK. [6]Dynamics of Intra-Cellular Organization, Institute Curie, PSL Research University, CNRS UMR144, Paris, France. [7]Present address: Biology Department, College of Science, Taibah University, Medina, Kingdom of Saudi Arabia. [8]Present address: Roche Pharmaceutical Research and Early Development, Pharmaceutical Sciences, Roche Innovation Center Basel, F. Hoffmann-La Roche Ltd., Basel, Switzerland. [9]Present address: Department of Biology, College of Sciences, Princess Nourah bint Abdulrahman University, Riyadh, Kingdom of Saudi Arabia. [10]Present address: Department of Anatomy, University Hospital Cologne, University of Cologne, Cologne, Germany. [11]These authors contributed equally: Martin Kollmar, Tobias Welz. ✉e-mail: mako@mpinat.mpg.de; Eugen.Kerkhoff@ukr.de

In a similar way, the long-range transport of RAB11 vesicles is mediated in mouse metaphase oocytes[9,10].

In both cases, the vesicle-originated actin filament tracks for long-range transport are generated by the cooperative activity of SPIRE actin nucleators and formin (FMN)-subgroup formins[9–11]. SPIRE proteins belong to the group of tandem Wiskott-Aldrich syndrome protein (WASP) homology 2 (WH2)-domain-based actin nucleators, that initiate actin filament assembly through WH2 actin-binding domains[12–14]. Actin nucleation factors of the SPIRE family (e.g. mammalian SPIRE1 and SPIRE2) coordinate actin filament assembly in cooperation with FMN-subgroup formins (e.g. mammalian FMN1 and FMN2) at organelle membranes[9–11]. SPIRE and FMN-subgroup formins interact with each other[15–18]. The interaction is mediated by a C-terminal formin SPIRE interaction (FSI) motif of FMN proteins and the N-terminal SPIRE kinase non-catalytic C-lobe domain (KIND) of SPIRE proteins. A model has been proposed in which SPIRE and FMN dissociate following actin filament nucleation allowing FMN to drive subsequent filament elongation through a processive association with the fast-growing filament barbed-end and recruitment of profilin-bound G-actin monomers through its FH2 and FH1 domains[14,19–21]. A C-terminal FYVE_2 (FYVE: FAB1, YOTB, VAC1, and EEA1) domain, consisting of the highly conserved SPIRE-box (SB), a FYVE-type zinc finger and C-terminal flanking sequences, mediates a vesicular localisation of *Drosophila melanogaster* and mammalian SPIRE proteins[14,15,22,23]. These C-terminal SPIRE sequences are homologous with RAB GTPase-binding domains of exocytic transport regulators such as SYTL3 (SYTL: synaptotagmin-like protein), RIM1 (RIM: RAB3-interacting molecule), Rabphilin-3A, SlaC2-a (SlaC2: synaptotagmin-like protein homologue lacking C2 domains) (MLPH: melanophilin), and SlaC2-cC (MYRIP: MYO7A and RAB interacting protein)[14,22,24]. The FYVE_2 domain of SPIRE1 interacts with the GTPases RAB27 and RAB3[11,25], which belong to the secretory group of the RAB GTPase vesicle transport regulators[26,27].

By forming a protein complex with MYO5, SPIRE coordinates actin filament track generation and motor protein activation[28]. A sequence motif in the central part of the mammalian SPIRE2 protein, which is conserved within the family of vertebrate SPIRE proteins, interacts with the globular tail domain (GTD) of the mammalian MYO5 motor proteins (globular tail domain binding motif, GTBM)[28]. RAB27A regulates the recruitment of SPIRE1 to melanin pigment-containing melanosomes in melanocytes, where SPIRE1, MYO5A, and FMN1 cooperate in the transport of mela-nosomes towards the plasma membrane[11]. In addition, SPIRE1, SPIRE2, FMN2 and MYO5B direct cortical transport of RAB11A vesicles in mouse oocytes[9,10]. The C-terminal globular tail domain of MYO5A links SPIRE2, MYO5A and RAB11A in a ternary complex by interacting with RAB11A and SPIRE2-GTBM-FYVE_2 proteins[28–30]. In neuronal networks a function of SPIRE, FMN, MYO5 and RAB11 proteins is associated with memory and learning[31–34].

To date, SPIRE proteins have not been described outside the animal kingdom and it is not known whether exocytic FMN/SPIRE/MYO5-directed actomyosin transport mechanisms exist in other eukaryotes and whether these mechanisms have evolved to meet the specific needs of the structural and signalling complexity of animal cells. Genomic studies show a discontinuous phylogenetic distribution of multicellularity, and the differences in cell biological mechanisms indicate that multicellularity evolved independently from single-celled ancestors in different multicellular eukaryotic lineages such as plants, fungi and animals[35–40]. Choanoflagellates are the closest unicellular relatives of animals[41–44]. These unicellular flagellate protists were first described in 1866 by James Clark, who also noted the strong structural resemblance between choanoflagellates and flagellated choanocytes of sponges, one of the earliest multicellular animals[42,45,46]. Thus, extant choanoflagellates are suggested to closely resemble the single-celled ancestors of the animal kingdom[37,40]. Studying choanoflagellates could therefore provide important insights into the cell biological mechanisms of animal origins[40]. The discovery of neuronal vesicle transport protein homologs in choanoflagellates[47], and the holozoan origin of many synaptic proteins suggests that a complex cell signalling machinery preceded animal evolution. Moreover, their development may have been a prerequisite for the evolution of the sophisticated diversity of animal cells and their neu-roendocrine communication mechanisms[48–50].

In this study, we identify SPIRE homologs in choanoflagellates, and the more distantly related unicellular protists called ichthyosporeans, and show that the organelle-associated cooperative function of SPIRE actin nucleators and MYO5 motor proteins precedes animal evolution. Choanoflagellate SPIRE efficiently initiates actin filament assembly and can compensate for mammalian SPIRE function in organelle transport. The major SPIRE protein interactions with FMN-subgroup formins, MYO5 and RAB GTPases are conserved between choanoflagellates and mammals. Consistently, we found choanoflagellate SPIRE and MYO5 proteins to colocalise at vesicular structures in the choanoflagellate *Salpingoeca rosetta*. Our findings strongly suggest that an actomyosin-based transport machinery, regulated by a network comprising RAB, SPIRE, FMN and MYO5 proteins, is conserved between choanoflagellates and animals and may have contributed to the evolution of multicellular animals by adding an extra degree of freedom to exocytic membrane transport systems.

## Results

### SPIRE proteins originated in the last common ancestor of Holozoa

Previous studies reveal that SPIRE proteins are conserved among mammals and insects, e.g. *Drosophila melanogaster*, and play an important role in organelle transport, including vesicle transport in oocytes and melanosome transport in melanocytes[9–11,22,51]. To determine the origin of SPIRE and its distribution in living species, we searched eukaryotic genomes and transcriptomes with TBLASTN[52], using the known SPIRE homologs from *Drosophila melanogaster* and humans as starting sequences.

The results of this analysis highlighted four main themes regarding the evolution, retention and duplication of SPIRE in eukaryotes (Fig. 1, Supplementary Fig. 1). First, SPIRE homologues exist in the majority of Holozoa, including multicellular animals and their closest unicellular relatives, but not in fungi or plants (Fig. 1b). Second, SPIRE is not universally conserved among Holozoa, e.g., SPIRE has been lost in the unicellular filasterean *Capsaspora owczarzaki* and in nematodes (Fig. 1b). Third, SPIRE duplication and loss have occurred independently several times during evolution. For example, in vertebrates, the ancient jawless fish (coastal hagfish *Eptatretus burgeri*) has a single SPIRE gene, while jawed vertebrates of more recent origin, such as birds and reptiles, contain three SPIRE genes (SPIRE1-3) (Fig. 1b, c; Supplementary Fig. 1). SPIRE3 then appears to have been independently lost in mammals and bony fish (euteleosts), leaving these groups with only SPIRE1 and SPIRE2 (Fig. 1b, Supplementary Fig. 1). Among invertebrates, SPIRE has been lost in ctenophores (comb jellies), placozoans (*Trichoplax adhaerens*, small disc-like animal), mesozoans (wormlike parasites of marine invertebrates), platyhelminthes (flat worms) and nematodes (round worms) (Fig. 1b). Gene duplicates, on the other hand, were only identified in the salmon louse *Lepeophtheirus salmonis* and the unicellular ichthyosporean *Amoebidium parasiticum*, which contains two SPIRE genes, one of which encodes a SPIRE-like protein lacking the RAB GTPase/membrane binding FYVE_2 zinc finger domain (Fig. 1c, Supplementary Fig. 1). Finally, SPIRE may have evolved in unicellular Holozoa by joining the KIND-WH2 actin assembly domains of an ancient SPIRE-like protein, such as that observed in the amoeba *Physarum poly-cephalum* (Fig. 1c), and the RAB GTPase/membrane binding FYVE_2 zinc finger of another unknown protein. Overall, these results suggest that SPIRE proteins play an important, but not essential, role in organelle transport in holozoans and that their biochemical activity, i.e. linking organelle membranes with actin assembly and myosin motors, is critical for these functions.

### Holozoan and metazoan SPIRE share a conserved domain organisation

A canonical animal SPIRE protein consists of the N-terminal KIND domain followed by two to four consecutive WH2 motifs, the GTBM in the central

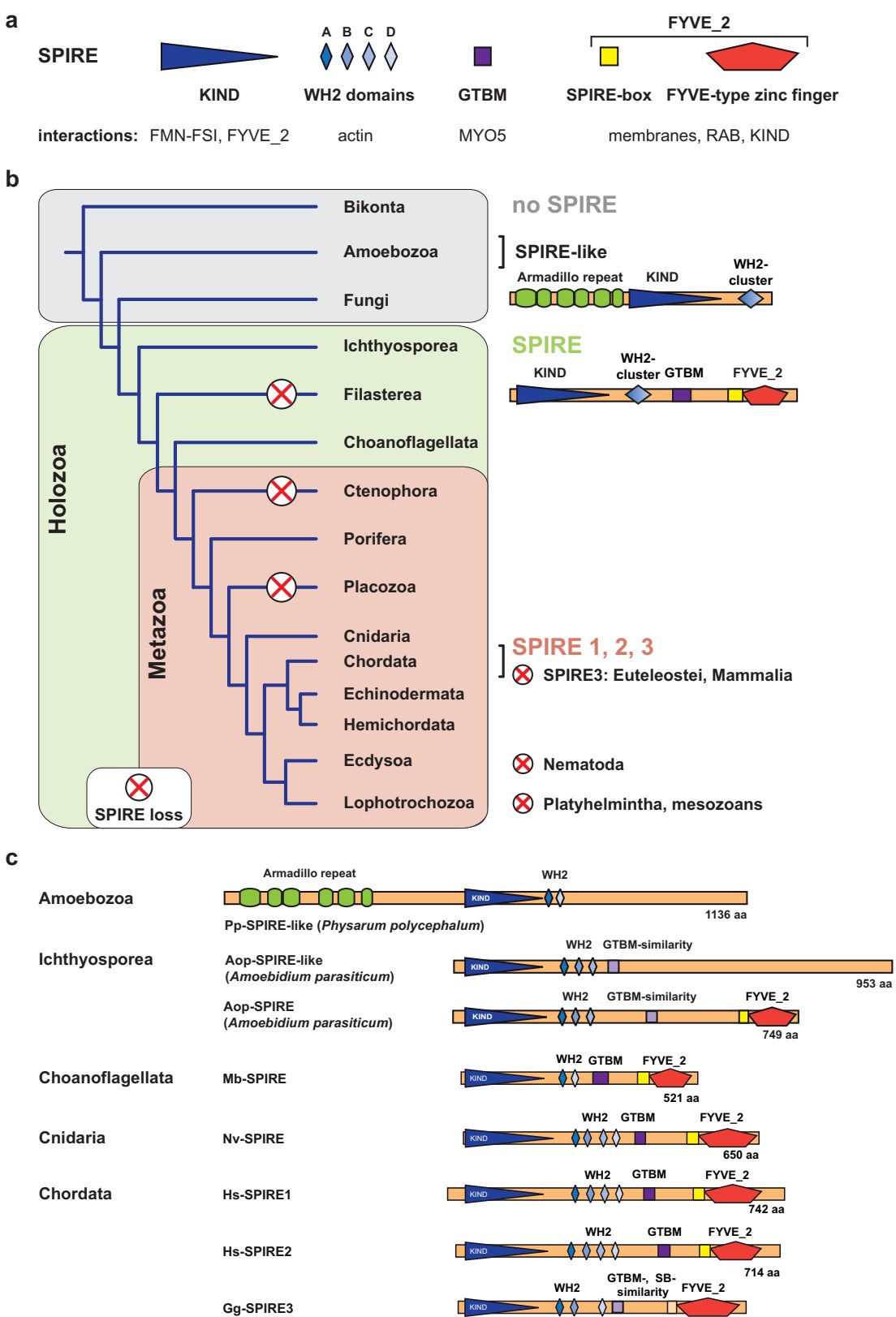

region and a tandem domain consisting of the SB motif and the FYVE-type zinc finger (FYVE_2 domain) (Fig. 1a). In contrast, the SPIRE-like protein of the amoebae *P. polycephalum* has a KIND domain and two WH2 motifs with strong homology to the other SPIRE proteins (Fig. 1b, c). However, it contains an N-terminal extension of 522 amino acids encoding armadillo repeats and lacks the GTBM and the C-terminal FYVE_2 domain, which are characteristic of SPIRE proteins (Fig. 1c).

The domain architectures of the SPIRE homologs of ichthyosporeans and choanoflagellates are identical to those of metazoans, except that these SPIRE proteins encode only two or three WH2 domains (Fig. 1c;

**Fig. 1 | The origin of SPIRE actin nucleator proteins precedes animal evolution. a** Overview of SPIRE protein domains and their specific interactions. The C-terminal FYVE_2 domain comprises the SPIRE-box and a FYVE-type zinc finger domain. Abbreviations: KIND kinase non-catalytic C-lobe domain, WH2 Wiskott-Aldrich syndrome protein homology 2, GTBM globular tail domain binding motif, FYVE after Fab 1/YOTB/Vac 1/EEA1, FSI formin SPIRE interaction motif, MYO5 class-5 myosin. **b** Origin and distribution of SPIRE and SPIRE-like proteins in living species. Genomic sequences and transcriptomes were screened with TBLASTN[52] using fruit fly and human SPIRE proteins as starting sequences. A schematic phylogenetic tree and allocated origin, distribution, duplications and losses of SPIRE (KIND, WH2, GTBM, FYVE_2) and SPIRE-like (KIND, WH2) proteins are shown.

SPIRE was not identified in Bikonta (including plants and SAR) and Fungi, but a SPIRE-like protein exists in Amoebozoa. The phylogenetic analysis suggests that SPIRE proteins evolved in single-celled Holozoa (animals and their closest single-celled relatives) through the conjunction of the KIND-WH2 actin assembly domains (SPIRE-like, as found in Amoebozoa) with the MYO5 binding motif (GTBM) and the membrane/RAB GTPase-binding FYVE_2 domain of unknown origins. Chordata have up to three SPIRE isoforms (SPIRE1, 2, 3). Red crosses indicate SPIRE losses. **c** The domain organization of SPIRE and SPIRE-like proteins is shown for extant species from Amoebozoa, Ichthyosporea, Choanoflagellata, Cnidaria and Chordata. Species abbreviations are: Mb *Monosiga brevicollis*, Nv *Nematostella vectensis*, Hs *Homo sapiens*, Gg *Gallus gallus*. aa amino acids.

Supplementary Fig. 2) and are distinct in their proposed MYO5 globular tail domain binding motif (GTBM) (Fig. 1c) (www.cymobase.org, SPIRE alignment)[53].

The structurally characterised mammalian SPIRE2-GTBM is a short 26 amino acids long sequence motif in the central part of the protein (Hs-SPIRE2, Arg402 - Glu427), which binds to the surface of the MYO5 globular tail domain[28]. The GTBM consists of an array of basic, hydrophobic and acidic residues, which are not structured in the sense that they fold into secondary structural elements[28]. Our alignment of holozoan SPIRE proteins identified in the central parts of the SPIRE proteins similar arrays in terms of their physicochemical properties, albeit with only low sequence conservation (www.cymobase.org, SPIRE alignment)[53].

The sequence homology of the choanoflagellate WH2 domains with individual domains of the mammalian tandem array of four WH2 domains (WH2-A, -B, -C, -D) suggests that the ancient holozoan SPIRE contained three or even four WH2 domains and that the present-day choanoflagellates retained single combinations of them (Fig. 1, Supplementary Fig. 2). The sauropsid (birds and reptiles) SPIRE3 homologs do not contain the third WH2 domain, or the sequence has been altered beyond recognition (Fig. 1c, chicken Gg-SPIRE3).

## Choanoflagellate SPIRE nucleates and depolymerises actin filaments

As an essential structural motif in the regulation of actin dynamics, the WH2 cluster is of exceptional importance for cell biological SPIRE functions[11,12,14]. Biochemical studies employing purified SPIRE WH2 domain tandem arrays derived from mammalian and fly SPIRE proteins, revealed actin nucleation, monomer sequestering and filament severing activities[12,19,54,55]. Actin nucleation was found to require an array of at least two WH2 domains, whereas filament severing and actin monomer sequestration activities require a single WH2 domain[55]. Interestingly, our comparative analysis of SPIRE protein sequences revealed that unicellular holozoans and vertebrates differ in containing arrays of 2-3 or 3-4 adjacent WH2 domains, respectively (Fig. 1c, Supplementary Fig. 2). To test the functional impact of this difference, we carried out in vitro actin filament assembly and disassembly assays using SPIRE from the choanoflagellate *Monosiga brevicollis* (Mb-SPIRE).

Mb-SPIRE contains two adjacent WH2 domains, which are homologous to the mammalian WH2-A and WH2-D domains (Fig. 1c; Supplementary Fig. 2). Recombinant choanoflagellate and mouse SNAP-tagged SPIRE proteins consisting of the N-terminal KIND domain and the WH2 domains (SNAP-Mb-SPIRE-KW; SNAP-Mm-SPIRE1-KW) were expressed and purified from *E. coli* (Supplementary Fig. 3), and subsequently analysed in pyrene actin assays to assess their biochemical properties. Comparable to Mm-SPIRE1-KW, Mb-SPIRE-KW accelerated actin filament assembly and disassembly in vitro (Figs. 2, 3). Specifically, in bulk pyrene assays analysing filament assembly from free actin monomers (2 μM, 5% pyrene-labelled), Mm-SPIRE1-KW induced assembly at concentrations above 5 nM and reached its maximal assembly activity at 50 nM (Fig. 2a). At higher concentrations (e.g. 2000 nM), presumably due to sequestration of monomers, Mm-SPIRE1-KW completely inhibited actin filament assembly. This is in line with previous results (Quinlan et al.[12]; Sitar et al.[55]). Consistently, in dilution-induced depolymerization assays, Mm-SPIRE1-KW effectively promoted disassembly of preformed actin filaments (Fig. 2b). Compared to Mm-SPIRE1-KW, Mb-SPIRE-KW was 20-fold less potent in actin assembly and reached its maximal activity at 1000 nM (Fig. 2c). Consistently, it was less effective in dilution-induced depolymerization assays (Fig. 2d). To experimentally test whether the large difference in activity between the two proteins was attributable, at least in part, to the difference in the number and type of WH2 domains in each protein, we generated a Mm-SPIRE1-KW mutant (Mm-SPIRE1-KW-ΔB,C) which lacks the internal WH2 domains B and C. As expected, Mm-SPIRE1-KW-ΔB,C was less potent (5-fold) than Mm-SPIRE-KW and reached its maximal actin assembly activity at 250 nM (Fig. 2e). This mutant was also less effective in promoting filament disassembly, and unexpectedly even appeared to stabilise filaments in depolymerisation assays at higher concentrations (Fig. 2f). The stabilisation could be due to actin filament binding by the mutant protein. The actin filament binding FAB domain of the *Dictyostelium discoedium* enabled/vasodilator-stimulated phosphoprotein (VASP) displays close similarity to actin monomer binding WH2 domains[56] and actin filament binding has been shown for the *Drosophila melanogaster* SPIRE proteins[21]. A potential actin filament binding activity of the SPIRE WH2 domains in general, as well as a reason why this activity might become dominant in the generated Mm-SPIRE1-KW-ΔB,C remains to be analysed in follow up studies.

To corroborate these data on the single filament level and to dissect the possible roles of SPIRE proteins in filament nucleation and elongation, we then employed total internal reflection fluorescence microscopy (TIRFM) assays. These assays revealed Mb-SPIRE-KW to nucleate actin filament assembly, albeit at a much lower frequency and at much higher protein concentrations as compared to Mm-SPIRE1-KW (Fig. 3a). At 90 s after initiation of the actin polymerisation reaction the number of generated actin filaments by Mm-SPIRE1-KW (10–250 nM) was 10-15-fold higher compared with the actin-only control (Fig. 3a, upper panel; Fig. 3b). In contrast, the nucleation activity of Mb-SPIRE-KW was barely visible at this time point (Fig. 3a, middle panel; Fig. 3c), and only became apparent at later time points (270 sec; Fig. 3a, lower panel; Fig. 3d). However, the actin filament elongation rates in the presence of Mb-SPIRE-KW and Mm-SPIRE1-KW proteins did not differ significantly from rates observed with actin control filaments indicating that both proteins nucleate rather than elongate actin filaments (Fig. 3e). Finally, we examined the disassembly of preformed actin filaments in the presence of Mm-SPIRE1-KW and Mm-SPIRE1-KW-ΔB,C. Increasing the concentration of Mm-SPIRE1-KW from 20 – 500 nM promoted rapid filament disassembly, with hardly any filaments remaining at the highest concentration as opposed to Mm-SPIRE1-KW-ΔB,C, which lacks two of the four WH2 domains (Supplementary Fig. 3c). Taken together these data show that the choanoflagellate Mb-SPIRE-KW protein can promote nucleation and disassembly of filaments just as its mammalian counterpart even though the activity of the choanoflagellate protein is considerably weaker in both cases. Our findings suggest that these differences are caused by the difference in the number of WH2 domains between Mm- and Mb-SPIRE-KW.

Another possibility is that the differences are due to differential binding of the choanoflagellate and mouse proteins to rabbit actin used in our

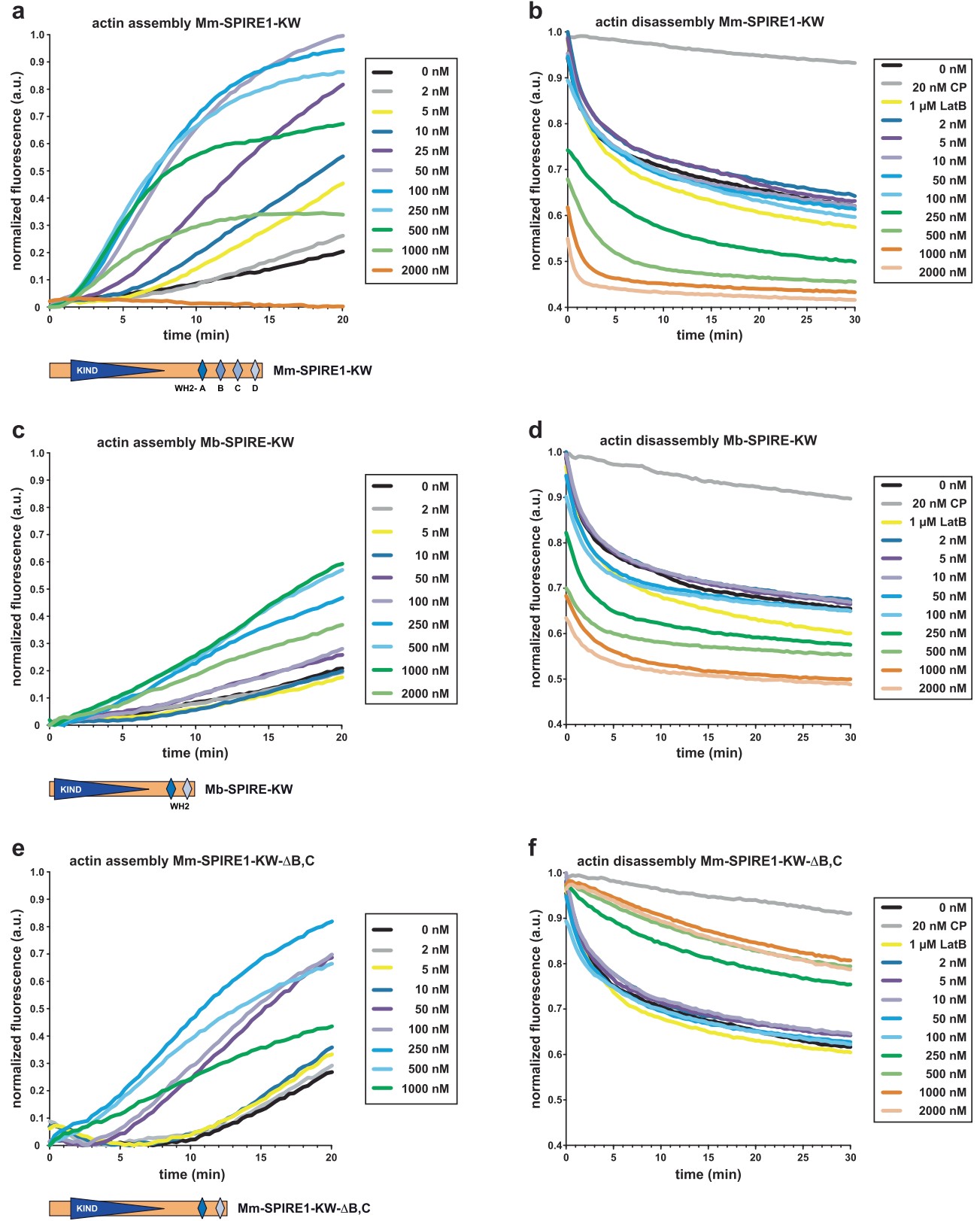

studies. However, sequence alignments show high conservation of vertebrate and choanoflagellate actin proteins (Supplementary Fig. 4a). The homology was also confirmed by a structural alignment of rabbit skeletal muscle actin (PDB-ID: 3MN5[57]) and an artificial intelligence (AI) generated protein structure prediction (ESMFold[58]) of the choanoflagellate *Monosiga* *brevicollis* actin protein (Supplementary Fig. 4b). All rabbit skeletal actin residues, which were found to interact in the *Drosophila melanogaster* SPIRE-WH2-D - rabbit skeletal actin complex structure (PDB-ID: 3MN5[57]) are conserved between rabbit skeletal actin and choanoflagellate actin (Supplementary Fig. 4a, b).

**Fig. 2 | Comparison of Mm-SPIRE1-KW and Mb-SPIRE-KW activities in actin assembly and disassembly. a, c, e** All tested SPIRE proteins promote actin assembly in pyrene-actin assays, but Mm-SPIRE1-KW containing four WH2 domains was most effective. G-actin (2 μM, 5% pyrene-labelled) was polymerised in 1x KMEI in presence and absence of Mm-SPIRE1-KW, deletion mutant Mm-SPIRE1-KW-ΔB,C and Mb-SPIRE-KW proteins at the concentrations as indicated. **b, d** Both Mm-SPIRE1-KW and Mb-SPIRE-KW accelerate dilution induced barbed-end depolymerization of preformed actin filaments suggesting sequestration of actin

monomers, but again, Mm-SPIRE1-KW was much more effective. **f** Interestingly, the mutant Mm-SPIRE1-KW-ΔB,C was not only weaker than the WT protein, but actually appeared to stabilize the depolymerizing filaments at higher concentrations, albeit clearly less potent than the heterodimeric capping protein (CP) which served as an internal control. For the assays, polymerised F-actin (0.5 μM in 1 x KMEI, 50% pyrene-labelled) was diluted to the critical concentration of 0.1 μM in 1 x KMEI and incubated in the presence and absence of the proteins and at the concentrations as indicated.

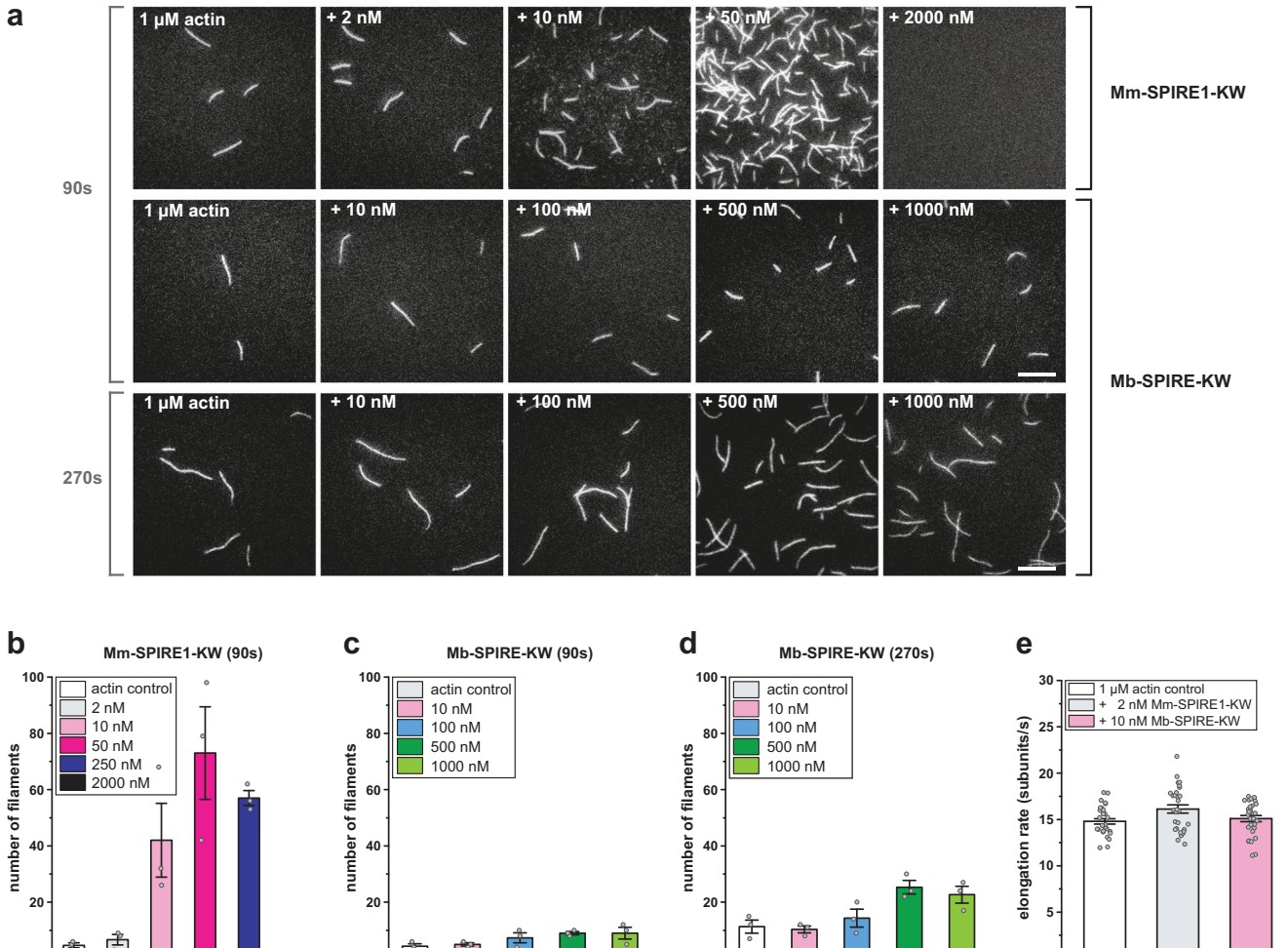

**Fig. 3 | Analysis of Mm-SPIRE1-KW and Mb-SPIRE-KW mediated actin assembly by TIRF imaging. a** TIRF microscopy images of actin assembly (1 μM, 20% ATTO488 labelled) in absence or presence of Mm-SPIRE1-KW or Mb-SPIRE-KW proteins at the concentrations as indicated after 90 s (upper and middle panel) and 270 s (lower panel). Scale bar represents 10 μm. As opposed to Mm-SPIRE1-KW, Mb-SPIRE-KW is a weak nucleator. **b–d** Quantification of nucleated filaments from assays as shown in (**a**). Filaments were counted 90 s and 270 s after initiation of

the polymerization reaction in an area of 50 μm x 50 μm. Bars represent mean ± SEM of three independent experiments. **e** Quantification of filament elongation rates from assays as shown in (**a**). Mm-SPIRE1-KW and Mb-SPIRE-KW did not noticeably affect actin filament elongation rates. At least 10 filaments were measured for each condition. Elongation rates are shown as mean ± SEM. The differences in speed of actin filament elongation were shown to be non-significant by the Kruskal-Wallis test.

## Major SPIRE protein interaction partners are conserved between choanoflagellates and mammals

Our sequence analysis suggests the origin of SPIRE proteins in the holozoan last common ancestor, which include animals and their closest single cell relatives[40,41] (Fig. 1, Supplementary Fig. 1). Mammalian SPIRE proteins interact with FMN-subgroup formins, MYO5 motor proteins and RAB GTPases[11,15,16,25,28]. The function of SPIRE proteins has been best characterised in mammalian organelle trafficking, where SPIRE proteins organise actomyosin force generation at organelle membranes including melanosomes in melanocytes and RAB11 vesicles in oocytes[9–11]. In melanocytes, SPIRE1 fulfils its transport function in a concerted action of

multiple proteins including the formin FMN1, the actin motor protein MYO5A and the small RAB GTPase RAB27A[11].

Homologues of the mammalian FMN-subgroup formin (FMN1, FMN2) and MYO5, have previously been identified in single-celled choanoflagellates[59,60]. The mammalian SPIRE1 interacting RAB27A and RAB3A GTPases are not encoded in choanoflagellates[11,25,27]. Genome studies on the evolution on RAB GTPases showed that the RAB27 and RAB3 regulators of the exocytic/secretory pathway originate in metazoa from a RAB8 GTPase gene, whose origin dates back to the last eukaryotic common ancestor (LECA)[27,61]. An interaction of mammalian SPIRE2 with RAB8 has been suggested by colocalisation experiments[62]. These observations suggest

that the organelle trafficking regulatory network comprising RAB, SPIRE, FMN and MYO5 (RSFM) proteins may operate in unicellular relatives of animals such as choanoflagellates and may have originated in a unicellular ancestor of animals.

In our attempt to answer the question of the origin of SPIRE functions, we have experimentally addressed whether major protein-protein interactions are conserved between mammalian and single-celled choanoflagellate SPIRE proteins. Specifically, we have explored whether the *Monosiga brevicollis* choanoflagellate SPIRE (Mb-SPIRE) protein interacts with FMN-subgroup formins, class-5 myosin motor proteins and RAB GTPases.

## The FMN-formin/SPIRE interaction is conserved between choanoflagellates and mammals

A hallmark of animal SPIRE proteins is their interaction with FMN-subgroup formins (*Drosophila* Cappuccino, mammalian FMN1 and FMN2)[15,16]. The cooperation of the two actin nucleators is mediated by an interaction of the SPIRE-KIND domain and the C-terminal formin SPIRE interaction (FSI) motif[16–18]. Indeed, the combination of an FSI motif (enriched in basic residues) and highly conserved formin homology domains (FH1, FH2) is the only obvious defining feature of FMN-subgroup formins[16,63] (Fig. 4a, b). Given this, and to probe the origin of the functional interaction between SPIRE and FMN proteins, we searched the genomes of unicellular organisms for genes encoding FH1 and FH2 domains and an FSI motif. This analysis revealed FMN-type formins in choanoflagellates (Fig. 4a, b). This is consistent with previous studies showing FMN-subgroup formins in *Salpingoeca rosetta* (*S. rosetta*) and *Monosiga brevicollis* (*M. brevicollis*) choanoflagellates[59]. These observations indicate that SPIRE and FMN-subgroup formins are of holozoan origin suggesting that their functions may have co-evolved[64].

Alignments of vertebrate FMN-FSI and choanoflagellate *Monosiga brevicollis* Mb-FMN-FSI protein sequences show that residues contacting the SPIRE-KIND domain in the human SPIRE1-KIND:FMN2-FSI complex are conserved (Fig. 4b)[17]. Acidic residues within the human SPIRE-KIND domain, which contribute to the SPIRE-FMN interaction[17,18] are also conserved between mammalian and choanoflagellate SPIRE proteins (Fig. 4c). A structural alignment revealed a high similarity between the crystallised human SPIRE1-KIND domain (PDB-ID: 2YLE[17]) and an AI generated structural model of the Mb-SPIRE-KIND domain, indicating that the surface presentation of acidic residues contacting the FMN-FSI motif is conserved (Fig. 4d).

To directly test whether the SPIRE/FMN interaction is conserved in choanoflagellate proteins we fused extended sequences of the putative choanoflagellate FSI motif with glutathione S-transferase (GST) (GST-Mb-FMN-eFSI) and performed GST-pulldown experiments with AcGFP1-Mb-SPIRE-KIND-WH2 (AcGFP1-Mb-SPIRE-KW) from lysates of transiently transfected human embryonic kidney 293 (HEK 293) cells. GST-Mb-FMN-eFSI interacted with AcGFP1-Mb-SPIRE-KW, but not with AcGFP1 alone (Fig. 4e). Similarly, GST alone did not interact with AcGFP1-Mb-SPIRE-KW (Fig. 4e). In addition, transient expression studies in human HeLa cells showed colocalisation of mRuby3-tagged C-terminal Mb-FMN (mRuby3-Mb-FMN-FH2-FSI) and AcGFP1-Mb-SPIRE proteins at vesicular structures (Fig. 4f). Colocalisation was not detected when the KIND and WH2 domains were deleted (AcGFP1-Mb-SPIRE-ΔKW) (Fig. 4f). Fluorescent proteins mRuby3 and AcGFP1 alone did not colocalise with AcGFP1-Mb-SPIRE-ΔKW or AcGFP1-Mb-SPIRE or mRuby3-Mb-FMN-FH2-FSI at vesicular structures (Fig. 4f). These data on interaction and colocalisation therefore suggest that the SPIRE/FMN interaction is conserved among choanoflagellates, insects and mammals.

## The choanoflagellate SPIRE protein is a RAB8 GTPase effector
Protein sequence alignment indicates that choanoflagellate SPIRE proteins, like their mammalian counterparts, contain C-terminal FYVE_2 domains that are similar to the FYVE_2 domains of mammalian RIM, Rabphilin-3A and melanophilin (MLPH) proteins (Fig. 5a). The FYVE_2 domains of these proteins mediate the interaction with RAB

GTPases[65]. In mammals, SPIRE1 interacts with RAB27A and RAB3A[11,25], and SPIRE2 may interact with RAB8[62], from which RAB27 and RAB3 originate[27]. The choanoflagellate genomes encode RAB8 (Fig. 5b) but not RAB27 and RAB3[27] suggesting that SPIRE/RAB interaction is conserved in choanoflagellate. To investigate this, we used AI to generate structural models of FYVE_2 domains from choanoflagellate and mouse SPIRE proteins (Mb-SPIRE-FYVE_2 and Mm-SPIRE1-FYVE_2) (ColabFold/AlphaFold2[66,67]) (Fig. 5c). Alignment of those models indicates that these proteins have almost identical structures (Fig. 5c), suggesting that interactions observed for mammalian SPIRE1-FYVE_2 are conserved in Mb-SPIRE-FYVE_2. Consistently, the comparison of an AI generated model of a potential choanoflagellate Mb-SPIRE-FYVE_2:Mb-RAB8 complex (ColabFold/AlphaFold-multimer[66–68]) with the X-ray crystallography resolved structure of the RAB27B:MLPH complex (PDB-ID: 2ZET,[69]) indicates that Mb-SPIRE FYVE_2 and MLPH FYVE_2 interact with RAB GTPases in a similar manner (Fig. 5d, e). In particular a major contact is the MLPH N-terminal SHD1 (synaptotagmin-like protein (SLP) homology domain 1) helix with the switch 2 region of RAB27B (Fig. 5e). Consistent with this the model of the Mb-SPIRE:Mb-RAB8 complex structure predicts that the N-terminal helix of the Mb-SPIRE-FYVE_2 domain links choanoflagellate SPIRE and RAB8 proteins.

To directly investigate the conservation of SPIRE/RAB interaction, we analysed the interaction of Mb-SPIRE with Mb-RAB8 by GST-pulldown assays. A recombinant GST-fusion protein with the Mb-RAB8-Q67L mutant (GST-Mb-RAB8-QL, GTP-locked form), pulled down transiently expressed AcGFP1-tagged Mb-SPIRE-ΔKW, but not AcGFP1 alone, from HEK 293 lysates (Fig. 5f). The GDP-locked RAB8-T22N protein fused to GST (GST-Mb-RAB8-T22N) did not pulldown AcGFP1-tagged Mb-SPIRE-ΔKW. We then further analysed the interaction of Mb-SPIRE and Mb-RAB8 by colocalisation studies in human HeLa cells. When transiently co-expressed mRuby3-tagged Mb-RAB8 (mRuby3-Mb-RAB8) and AcGFP1-tagged Mb-SPIRE-ΔKW (AcGFP1-Mb-SPIRE-ΔKW) strongly colocalised (Fig. 5g). Neither mRuby3- nor AcGFP1 alone co-localised at vesicular structures in the absence of AcGFP1-Mb-SPIRE-ΔKW and Mb-RAB8 (Fig. 5g). The prominent colocalisation of mRuby3-Mb-RAB8 and AcGFP1-Mb-SPIRE-ΔKW, further supported the protein complex formation of Mb-RAB8 and Mb-SPIRE (Fig. 5g). Considering the high degree of conservation of vertebrate RAB8 function in vesicle transport processes towards the cell membrane[27,70,71], the choanoflagellate SPIRE/RAB8 interaction shown here suggests that SPIRE proteins originated as part of the exocytic transport machinery.

## The MYO5/SPIRE interaction is conserved between choanoflagellates and mammals
Mammalian SPIRE2 interacts with class-5 myosin motor proteins via direct contact of a centrally located and conserved GTBM motif with the globular tail domains of the MYO5A, MYO5B and MYO5C motor proteins[28]. The choanoflagellate *Monosiga brevicollis* Mb-MYO5 protein displays high similarity to mammalian MYO5s (Fig. 6a) and an AI generated structure model (ColabFold/AlphaFold-multimer[66–68]) shows a perfect alignment of the predicted Mb-MYO5-GTD structure with the crystal structure of the human MYO5A-GTD (PDB-ID: 5JCY[28]) (Fig. 6b). In contrast, the sequence alignment did not reveal a GTBM-related motif in choanoflagellate SPIRE proteins (www.cymobase.org, SPIRE alignment)[53].

To determine whether the SPIRE/MYO5 interaction, and thus coordinated actin nucleation and motor protein activation is conserved between choanoflagellates and mammals, we tested the interaction of the *M. brevicollis* Mb-SPIRE and Mb-MYO5 proteins. In GST-pulldown assays, GST-Mb-MYO5-GTD strongly interacted with AcGFP1-tagged full-length Mb-SPIRE (AcGFP1-Mb-SPIRE), transiently expressed in HEK 293 cells (Fig. 6c). The AcGFP1-Mb-SPIRE protein on the other hand did not interact with GST nor did the AcGFP1 protein interact with GST-Mb-MYO5-GTD (Fig. 6c). Further support of interaction was obtained by transiently co-expressing choanoflagellate Mb-SPIRE and Mb-MYO5

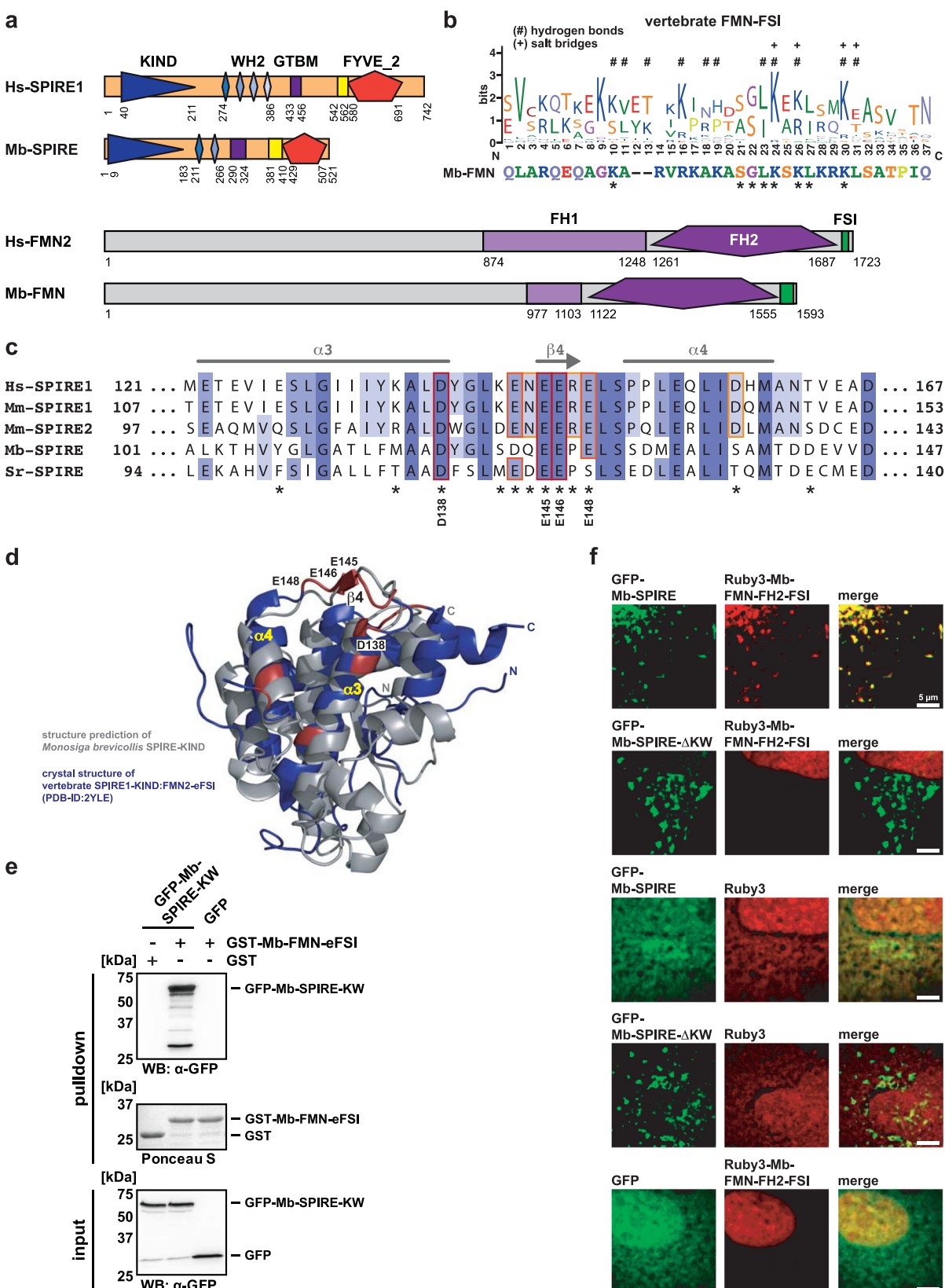

proteins in human HeLa cells. The choanoflagellate N-terminal deletion mutant AcGFP1-Mb-SPIRE-ΔKW colocalised with mRuby3 tagged Mb-MYO5-GTD at vesicular structures (Fig. 6d). By generating specific deletion mutants of the AcGFP1-Mb-SPIRE protein, we mapped the Mb-MYO5 interaction sequence to a stretch of 36 amino acids located in the central

region of the Mb-SPIRE protein, between the WH2 domains and the SPIRE-box, which is exactly the region where the MYO5 globular tail domain binding motif (GTBM) of mammalian SPIRE proteins is located (Fig. 6e, f; Fig. 1c). For the *Salpingoeca rosetta* Sr-SPIRE and Sr-MYO5 proteins, the SPIRE MYO5 binding motif was also mapped to the central

**Fig. 4 | SPIRE interactions with FMN-subgroup formins are conserved across distant. a** Domain organisation of human (Hs *Homo sapiens*) and *Monosiga brevicollis* (Mb) SPIRE proteins and formin (FMN) subfamily formins. SPIRE proteins share KIND (kinase non-catalytic C-lobe domain), WH2 (Wiskott-Aldrich syndrome protein homology 2) domains, GTBM (globular tail domain binding motif) and FYVE_2 domain (after Fab1/YOTB/Vac1/EEA1). FMN proteins share the formin homology domains (FH1 and FH2), the C-terminal formin-SPIRE interaction sequence (FSI) and as-yet uncharacterised large N-terminus. Protein structures are drawn to scale and numbers indicate amino acid residues. **b** WebLogo[108,109] showing amino acid conservation within vertebrate FMN-FSI protein sequences. Residues forming hydrogen bonds (#) and salt bridges (+) with the SPIRE-KIND are labeled respectively[17]. Below the corresponding C-terminal sequence part of Mb-FMN is depicted, adjusted to the WebLogo and conserved residues are labeled (*). **c** Multiple sequence alignment of vertebrate (Hs *Homo sapiens*, Mm *Mus musculus*) and choanoflagellate (Mb *Monosiga brevicollis*, Sr *Salpingoeca rosetta*) SPIRE proteins. The acidic cluster of the SPIRE KIND domain is shown spanning α-helices α3 and α4 as well as the β-sheet β4[17]. Coloring indicates amino acid conservation at specific positions ranging from high (dark blue) to low (light blue) conservation. Asterisks indicate contact sites between vertebrate SPIRE1-KIND and FMN2-FSI sequences[17]. Contact sites conserved in *Monosiga brevicollis* are labelled accordingly. Numbering indicates amino acid residues. **d** Protein structure alignment of the ESMFold[58] predicted Mb-SPIRE KIND domain (grey) and the crystal structure of the vertebrate SPIRE1-KIND:FMN2-FSI complex (PDB-ID: 2YLE, blue[17]). SPIRE1-KIND amino acid residues forming contact sites with FMN2-FSI are colored in red. Contact sites which are conserved in *Monosiga brevicollis* are labelled accordingly (compare with (**c**)). The α-helices α3 and α4 as wells as β-sheet β4 are indicated. The FSI peptide structure is hidden for clarity. **e** GST-pulldown assay with purified GST-Mb-FMN-eFSI and lysates from HEK 293 cells transiently expressing N-terminal (KIND and WH2) AcGFP1-tagged Mb-SPIRE (GFP-Mb-SPIRE-KW, input). GST and AcGFP1 (GFP) were used as controls and Ponceau S staining shows equal amounts of GST and GST-tagged proteins. *N* = 2 experimental repeats. **f** Localisation of transiently co-expressed tagged full-length Mb-SPIRE and Mb-SPIRE-ΔKW (AcGFP1; GFP-Mb-SPIRE, GFP-Mb-SPIRE-ΔKW; green) and Mb-FMN-FH2-FSI (mRuby3; Ruby3-Mb-FMN-FH2-FSI; red) in human HeLa cells was analysed by fluorescence microscopy. AcGFP1 (GFP) and mRuby3 (Ruby3) expressions were used as controls. Deconvoluted images indicate the localisation of Mb-SPIRE and Mb-SPIRE-ΔKW proteins on vesicular structures. Mb-FMN-FH2-FSI colocalises with Mb-SPIRE, but not with Mb-SPIRE-ΔKW. Scale bars represent 5 μm. At least six cells from two distinct experiments were imaged for each condition and the cytoplasmic region of one representative cell is presented here.

region of the Sr-SPIRE whose protein sequence is very similar to the Mb-SPIRE-GTBM (Supplementary Fig. 5).

Comparison of the choanoflagellate Mb/Sr-SPIRE MYO5 interaction motifs with vertebrate SPIRE GTBMs shows only low sequence homology (Fig. 6g). However, the array of basic (blue), hydrophobic (green) and acidic residues (red) in the vertebrate GTBMs appears to be conserved in the Mb-SPIRE GTBM (Fig. 6g). Comparing their sequences, the Mb-MYO5 globular tail domain is closely related to the vertebrate MYO5-GTDs[29,60]. The AI predicted structure of Mb-MYO5-GTD in complex with the experimentally minimised Mb-SPIRE-GTBM (Fig. 6b) suggests that the SPIRE binding site is conserved for the human and *Monosiga brevicollis* MYO5. The structure model suggests that the N-terminal part of the Mb-SPIRE-GTBM and Hs-SPIRE2-GTBM bind the corresponding MYO5-GTD in a similar way, however, the C-terminal parts of the SPIRE-GTBM peptides interact with the myosin differently. Nevertheless, the interaction of Mb-SPIRE and Mb-MYO5 indicates that the cooperation of actin nucleation and motor protein activation is conserved between choanoflagellates and mammals.

In summary, our GST-pulldowns and colocalisation studies in human HeLa cells showed that major features such as MYO5 and FMN-subgroup formin interactions and secretory group RABs interactions are conserved between mammals and choanoflagellates. The vesicular localisation of the choanoflagellate Mb-SPIRE in human HeLa cells further shows that the choanoflagellate SPIRE protein is targeted to organelle membranes, as was previously found for its mammalian homologues[9–11,22,23]. These observations suggest that the functional relationship between these proteins in regulating membrane trafficking and the cytoskeleton is also conserved (Fig. 7a).

### Vesicular colocalisation of SPIRE and MYO5 in the choanoflagellate *Salpingoeca rosetta*

Recent protocols have established choanoflagellates as a model organism for molecular cell biology studies[72–74]. This enabled us to monitor the subcellular localisation of fluorescently tagged recombinant choanoflagellate Sr-SPIRE and Sr-MYO5 proteins directly in live *S. rosetta* by transient overexpression. Vectors for transgene expression were generated, encoding an mScarlet-I-tagged full-length Sr-SPIRE protein (mScarlet-Sr-SPIRE) and an mNeonGreen-tagged C-terminal Sr-MYO5 protein encoding the coiled-coil (cc) region and the globular tail domain (mNeonGreen-Sr-MYO5-cc-GTD). The encoded inserts were flanked by non-coding sequences upstream and downstream of the *Salpingoeca rosetta* non-muscle actin coding region[72]. The MYO5-cc-GTD protein encompasses the coiled-coil structural motifs for dimerisation and the globular tail domain for cargo interaction and thereby encodes all sequences previously recognised to be required for efficient cargo targeting[30,62]. Co-expression of mScarlet-Sr-SPIRE and mNeonGreen-Sr-MYO5-cc-GTD proteins in *S. rosetta* revealed cocolocalisation of the proteins at vesicular structures including larger structures adjacent to the apical pole of cells (Fig. 7b). This suggests that the cooperation of SPIRE and MYO5 functions in providing forces for actomyosin-based organelle transport is conserved in choanoflagellates and may promote transport towards the apical pole.

### Choanoflagellate Mb-SPIRE rescues mouse SPIRE function in melanosome transport

To test whether the SPIRE function in actin filament assembly at organelle membranes is conserved throughout evolution, we finally analysed whether the Mb-SPIRE protein can rescue SPIRE1/2 function in mouse melanocytes. Previously, we found that in melanocytes SPIRE1/2 and FMN1 proteins disperse melanosomes into the peripheral cytoplasm of melanocytes by generating an actin filament network used by MYO5A to transport melanosomes[11]. Rescue of perinuclear melanosome clustering seen in SPIRE1/2 siRNA depleted melanocytes (Fig. 8a) by expression of siRNA resistant SPIRE thus provides a convenient read-out of activity. Previous siRNA studies have shown that expression of SPIRE1/2 was reduced to 22.0% and 18.7%, respectively, as compared with control non-targeting siRNA[11]. Here we found that transgenic expression of eGFP-Mb-SPIRE and mammalian eGFP-tagged SPIRE1 and eGFP-tagged SPIRE2, but not eGFP alone, rescued the perinuclear melanosome clustering in SPIRE1/SPIRE2-depleted mouse melanocytes (Fig. 8b, c). Interestingly, we observed that in a subset of cells, low expression of Mb-SPIRE caused a hyper-dispersion phenotype, similar to that seen in experiments with SPIRE2 (Fig. 8b, c)[11]. These findings suggest that Mb-SPIRE interacts with endogenous FMN1 in melanocytes to generate local actin filament tracks for melanosome dispersion. In line with this notion we found an interaction of choanoflagellate Mb-SPIRE-KIND with the mouse FMN1-FH2-FSI protein in GST-pulldown assays (Fig. 8d). The mechanistic basis of the hyper-dispersion phenotype remains unclear at present, but may be due to a different RAB GTPase specificity of SPIRE1 versus SPIRE2 and Mb-SPIRE (see below). Nevertheless, altogether these data indicate that the mechanism of action of SPIRE proteins in transport is conserved between multicellular animals and their closest unicellular relatives.

### Discussion
Mammalian SPIRE proteins organise actin filaments at organelles and cell membranes[14]. In this way, the actin nucleators were described to contribute to the generation of forces driving vesicle transport, factor externalisation, pronuclear migration and mitochondrial fission[9–11,75–77]. The role of mammalian SPIRE proteins in exocytic organelle transport processes has been studied in detail[9–11]. By interactions with FMN formins and MYO5 myosin motor proteins mammalian SPIRE1 and SPIRE2 proteins coordinate actin filament generation and MYO5 motor activation[16–18,28]. The combined action of actin filament nucleators, elongators and myosin motors at vesicle membranes generates actin meshworks, which serve as tracks for MYO5 motor protein mediated centrifugal dispersion of the vesicles[10,11].

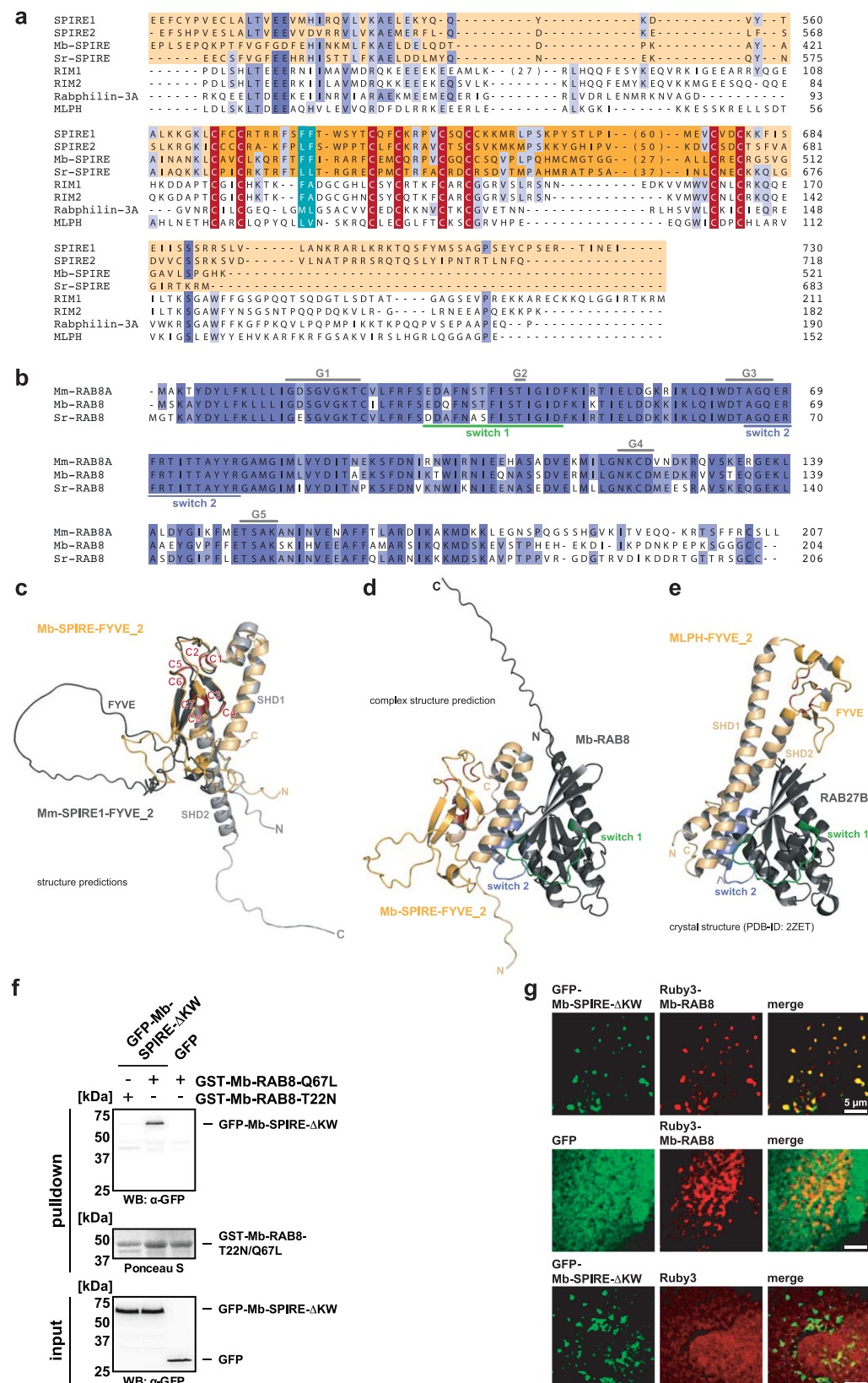

To date, SPIRE has been studied in insects, tunicates and vertebrates[14,51,78–80], limiting our knowledge on the evolution of the protein. Our comprehensive genome analysis of extant species identified SPIRE proteins outside metazoans in ichthyosporeans and choanoflagellates, which are the closest unicellular relatives of animals. We have characterised choanoflagellate SPIRE proteins using biochemical assays and in vivo localisation studies. The choanoflagellate SPIRE tandem WH2 actin binding domains efficiently nucleate actin filament assembly and choanoflagellate SPIRE and MYO5 proteins colocalise at vesicular structures in choanoflagellates. Together these

**Fig. 5 | Choanoflagellate SPIRE interacts with RAB8. a** Multiple protein sequence alignment of the FYVE_2 domains of the indicated exocytic proteins. Blue coloring indicates amino acid conservation at specific positions ranging from high (dark blue) to low (light blue) conservation. The eight cysteines for $Zn^{2+}$ ion binding and the hydrophobic turret loop are labelled in red and cyan, respectively. SPIRE protein sequences corresponding to the synaptotagmin-like protein (SLP) homology domains SHD1 and SHD2 are highlighted in light orange, sequences corresponding to the FYVE domain are highlighted in orange (consistent with the coloring in panels **c–e**). Large loop sequences within SPIRE and RIM1 proteins are hidden for clarity and shown in brackets. Numbering indicates amino acid residues. **b** Multiple protein sequence alignment of RAB8 proteins from *Mus musculus* (Mm), *Monosiga brevicollis* (Mb) and *Salpingoeca rosetta* (Sr). Coloring indicates amino acid conservation at specific positions ranging from high (dark blue) to low (light blue) amino acid conservation at specific positions. The highly conserved G motifs (G1 to G5) are indicated, as well as the residues forming switch 1 (green) and switch 2 (blue) regions. Numbering indicates amino acid residues. **c** Protein structure alignment of the predicted (ColabFold/AlphaFold2[66,67]) FYVE_2 domains of Mb-SPIRE (orange) and Mm-SPIRE1 (grey). Indicated are the synaptotagmin-like protein homology domains (SHD1, SHD2), the FYVE-type zinc finger and the $Zn^{2+}$ ion binding cysteines (C1-C8, labelled in red). **d** Predicted protein complex (ColabFold/

AlphaFold-multimer[66–68]) formed by *Monosiga brevicollis* Mb-SPIRE-FYVE_2 (orange) and Mb-RAB8 (grey). The switch 1 (green) and switch 2 (blue) interaction surfaces of RAB8 are indicated. The $Zn^{2+}$ ion binding cysteines of Mb-SPIRE-FYVE_2 are colored in red. **e** Crystal structure of the protein complex formed by the melanophilin (MLPH, SlaC2-a) FYVE_2 domain and RAB27B (PDB-ID: 2ZET[69]) is shown. SHD1, SHD2 and FYVE-type zinc finger of MLPH as well as switch 1 and switch 2 regions of RAB27B are indicated. **f** GST-pulldown assay with purified GTP-locked GST-Mb-RAB8-Q67L and GDP-locked GST-Mb-RAB8-T22N proteins, respectively, from lysates of HEK 293 cells transiently expressing C-terminal AcGFP1-tagged Mb-SPIRE (GFP-Mb-SPIRE-ΔKW, input). AcGFP1 (GFP) was used as negative control and Ponceau S staining shows equal amounts of GST-tagged proteins. *N* = 2 experimental repeats. **g** Localisation of transiently co-expressed tagged C-terminal Mb-SPIRE (AcGFP1; GFP-Mb-SPIRE-ΔKW; green) and Mb-RAB8 (mRuby3; Ruby3-Mb-RAB8; red) in human HeLa cells was analysed by fluorescence microscopy. AcGFP1 (GFP) and mRuby3 (Ruby3) expressing cells were used as controls. Deconvoluted images indicate the localisation of the proteins on vesicular structures. Scale bars represent 5 μm. At least six cells from two distinct experiments were imaged for each condition and the cytoplasmic region of one representative cell is presented here.

data indicate that vesicular SPIRE actomyosin functions precede animal evolution.

We propose that at the stage of Holozoa a previously established SPIRE-like KIND-WH2-dependent actin nucleation machinery (already present in Amoebozoa) fused with motifs for binding membranes, RAB GTPases and MYO5 motor proteins. This has linked actin filament generation and myosin actin motor protein activity at membranes and introduced actomyosin transport mechanisms[14]. Membrane trafficking has a central role in creating the diversity of eukaryotic cellular architectures and communication throughout evolution[61,81]. The transition to multicellularity in animals faced an increased need for cargo sorting and trafficking to coordinate the structural diversity and the complex intercellular signalling networks of the distinct cell-types[61]. This is reflected by an increase of trafficking routes, which requires a parallel amplification of the regulatory molecular machinery. An evolution of the SPIRE/MYO5 actomyosin transport pathway in the unicellular ancestors of the animals may thereby be considered as a prerequisite of animal evolution by contributing to the extended trafficking needs of multicellular animals.

Choanoflagellates are the closest unicellular relatives of animals[41–44]. The unicellular organisms have a very complex life cycle including facultative multicellularity. *Salpingoeca rosetta* choanoflagellates can transition from free-swimming unicellular states to multicellular chain or rosette colonies[82,83]. More recently the newly identified choanoflagellate species *Choanoeca flexa* was found to form multicellular cup-shaped colonies, which exhibit light-regulated collective contractility[84–86]. This indicates that the cells of the choanoflagellate colonies communicate with each other and coordinate their actions, in response to external stimuli, which are basic requirements of animal life. The mode of communication as well as a role of actomyosin organelle transport in the cell communication is unknown. Thus, studying the role of SPIRE/MYO5 transport mechanism in light of choanoflagellate colony formation and behaviour will help evaluate its role in the origin of animal multicellularity.

As an important achievement of our work, we have identified choanoflagellate SPIRE as a RAB8 GTPase interacting protein. A functional relation of SPIRE, RAB8 and MYO5, as we have shown here for choanoflagellates, has also been found in mammalian cells, where the mammalian SPIRE2-GTBM-FYVE_2, RAB8A and MYO5A-cc-GTD proteins colocalise at vesicular structures in human HeLa cells[62]. In eukaryotic cells, RAB8 acts in Golgi to plasma membrane transport[27]. The holozoan RAB8 gene has a large set of duplications in metazoans, giving rise, among others, to the RAB3 and RAB27 GTPases in animals[27]. Interactions of the mammalian SPIRE1 protein with RAB3 and RAB27 GTPases have been described[11,25]. Yet, it remains to be analysed if the choanoflagellate SPIRE - RAB8 interaction is conserved throughout animal evolution. RAB8, RAB27 and RAB3 GTPases are members of the secretion group of RAB GTPases and regulate

transport towards the cell membrane[27]. RAB27 is considered to operate in regulated secretion of dense core vesicles[24,87] and RAB3 functions in $Ca^{2+}$-triggered exocytosis of synaptic vesicles[88]. RAB8 function has been best studied in apical protein transport in epithelial cells[70,71,89] and contributes to a constitutive secretion to replenish material at the apical membrane. Gene knockout experiments in mice revealed a critical role of the mammalian RAB8A and RAB8B in the transport of apical proteins in epithelial cells[70,71]. The mammalian MYO5B and RAB11A proteins, which function together with SPIRE1 and SPIRE2 in ooplasmic actomyosin long-range vesicle transport[10], also play a role in apical transport processes[89]. Epithelial cells of the digestive tract are a major site of *SPIRE2* gene expression[90]. Combined these data suggest a role of SPIRE2 in apical transport processes. However, a function of SPIRE2 in the regulation of epithelial polarity has not yet been explored.

Here we have discovered localisation of choanoflagellate MYO5 at larger apical vesicles in choanoflagellate cells. Similar to epithelial cells, choanoflagellates display a polarised structure, which is characterised by an apical actin-rich microvillar collar surrounding an apical flagellum[83,91,92]. The mechanistic principles of the role of the MYO5 motor protein in establishing apical polarity may therefore be conserved between choanoflagellates and animal epithelial cells. Our data on choanoflagellate SPIRE/MYO5 vesicular colocalisation, the interaction of choanoflagellate SPIRE with RAB8, the vesicular colocalisation of mammlian SPIRE2-GTBM-FYVE_2, MYO5A-cc-GTD and RAB8A[62] and the high epithelial expression of the mouse *SPIRE2* gene[90] all support the idea that the SPIRE proteins may be involved in apical transport processes. Thus, in future work, gene knockout experiments of *MYO5*, *SPIRE* or *RAB8* in choanoflagellates may provide interesting insights into the regulatory mechanisms of apical polarity.

SPIRE/FMN-regulated actin structures contribute to mammalian oocyte maturation and fertilisation[9,10,75]. During mouse oocytes maturation SPIRE1 and SPIRE2 cooperate with FMN2 in regulating actin structures required for chromosome segregation and polar body formation[9]. During fertilisation, mouse SPIRE2 and FMN2 cooperate in the assembly of actin structures contributing to the inward movement of the male pronucleus inside mouse zygotes[75]. Our in vitro actin polymerisation and protein interaction studies show that SPIRE actin nucleation activity and the interaction with FMN-subgroup formins precede animal evolution. We therefore assume a general function of SPIRE/FMN cooperation in animal reproduction to be likely. It will be important for the understanding of the basic mechanistic principles of animal reproduction to address the roles of SPIRE and the FMN-subgroup formins in oocyte maturation and fertilisation processes of early branching animals such as sponges or sea anemones.

Our combined genome and protein interaction studies indicate that a SPIRE/MYO5/FMN actin nucleator/myosin motor protein complex

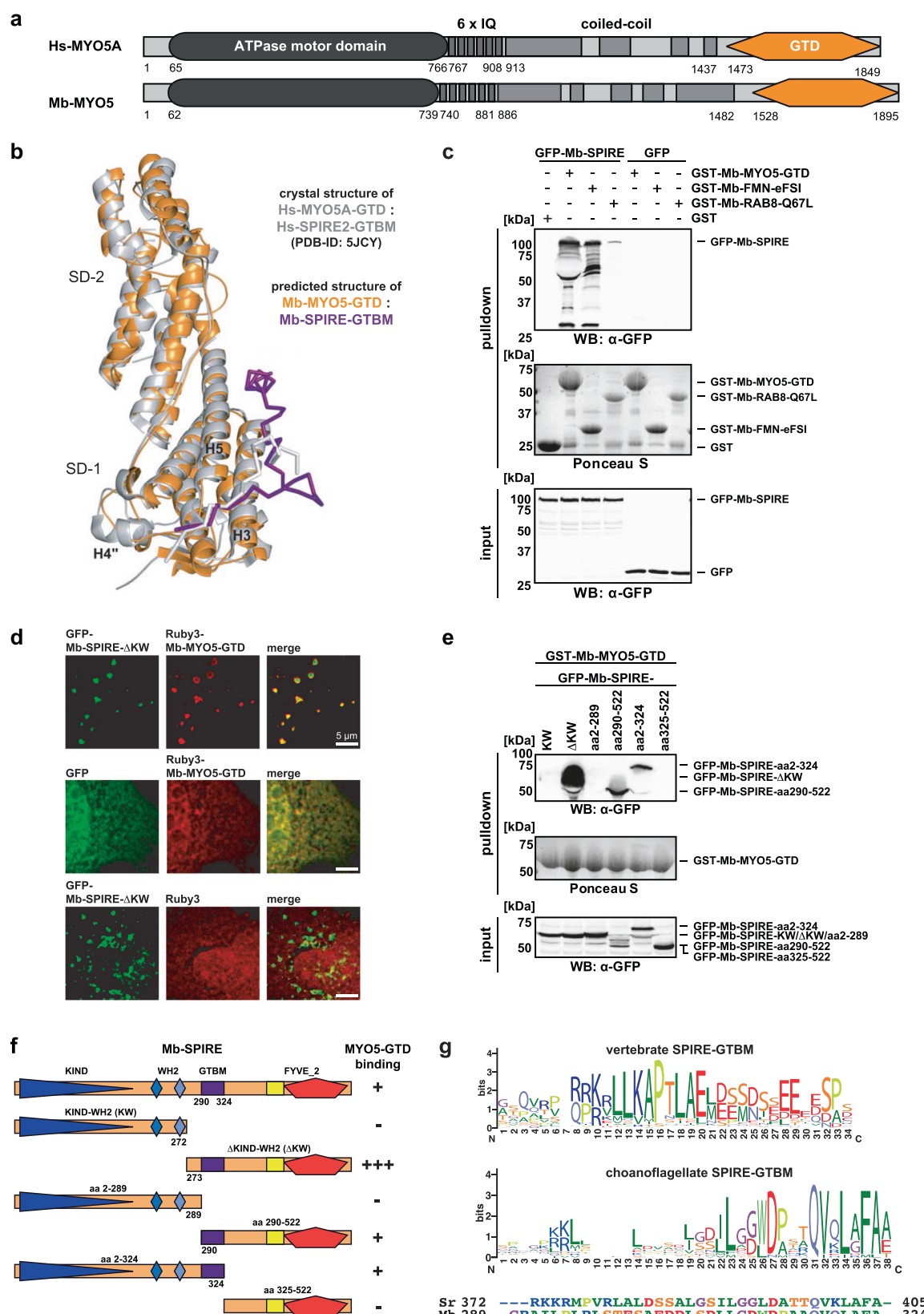

originated in the unicellular ancestors of animals and is associated to exocytic organelle transport via RAB8, RAB3 and RAB27 interactions. The protein complex may have contributed to the evolution of animals and their sophisticated networks of polarised and communicating cells by providing an extended toolkit of actin and myosin functions in intracellular transport processes. The importance of SPIRE's functions is supported by its high level of retention throughout the animal kingdom. Studying the cell biology of SPIRE proteins in single-celled holozoans and early branching animals will be a rewarding venture in our aim to understand the basic molecular mechanism underlying animal life.

**Fig. 6 | Choanoflagellate SPIRE interacts with myosin 5. a** Domain organisation of human (Hs, *Homo sapiens*) and *Monosiga brevicollis* (Mb) myosin-5 (MYO5) proteins. The MYO5 proteins share all characterised functional domains, including the ATPase motor domain, calmodulin-binding IQ motifs, coiled-coil regions and a C-terminal cargo-binding globular tail domain (GTD). Numbers indicate amino acid residues. **b** Protein structure alignment of the ColabFold/AlphaFold-multimer[66–68] predicted Mb-MYO5 globular tail domain (GTD, orange) in complex with Mb-SPIRE-GTBM (purple) and the crystal structure of the human MYO5A-GTD in complex with the human SPIRE2-GTBM (PDB-ID: 5JCY, grey/white[28]). GTD subdomains (SD-1, SD-2) are indicated as well as the helices H3, H4" and H5. **c** GST-pulldown assay with purified GST-Mb-MYO5-GTD, GST-Mb-FMN-eFSI and GTP-locked GST-Mb-RAB8-Q67L from lysates of HEK 293 cells transiently expressing full-length AcGFP1-tagged Mb-SPIRE (GFP-Mb-SPIRE, input). GST and AcGFP1 (GFP) were used as controls and Ponceau S staining shows equal amounts of GST and GST-tagged proteins. *N* = 2 experimental repeats. **d** Localisation of transiently co-expressed tagged C-terminal Mb-SPIRE (AcGFP1; GFP-Mb-SPIRE-ΔKW; green) and Mb-MYO5-GTD (mRuby3; Ruby3-Mb-MYO5-GTD; red) in human HeLa cells was analysed by fluorescence microscopy. AcGFP1 (GFP)

and mRuby3 (Ruby3) expressions were used as controls. Deconvoluted images indicate the colocalisation of the mRuby3-Mb-RAB8 and AcGFP1-Mb-SPIRE-ΔKW proteins on vesicular structures. Scale bars represent 5 μm. At least six cells from two distinct experiments were imaged for each condition and the cytoplasmic region of one representative cell is presented here. **e** GST-pulldown assay with purified GST-Mb-MYO5-GTD from HEK 293 cell lysates transiently expressing different N-terminal and C-terminal AcGFP1(GFP)-tagged Mb-SPIRE protein fragments. Ponceau S staining shows equal amounts of GST-Mb-MYO5-GTD proteins. *N* = 2 experimental repeats. KW, KIND-WH2. Numbering indicates amino acid residues. **f** Schematic representation of N-terminal and C-terminal Mb-SPIRE protein fragments as used in (**e**) and their ability to bind (+) or not to bind (-) to Mb-MYO5-GTD. GTBM, globular-tail-domain-binding motif. **g** WebLogos[108,109] depicting the amino acid conservation within vertebrate (upper panel, 93 sequences) and choanoflagellate (lower panel, 18 sequences) SPIRE GTBM amino acid sequences. Corresponding amino acids of *Monosiga brevicollis* (Mb) and *Salpingoeca rosetta* (Sr) SPIRE proteins experimentally shown to be involved in MYO5-GTD binding are depicted below and aligned to the choanoflagellate WebLogo, which is provided with respective amino acid boundaries.

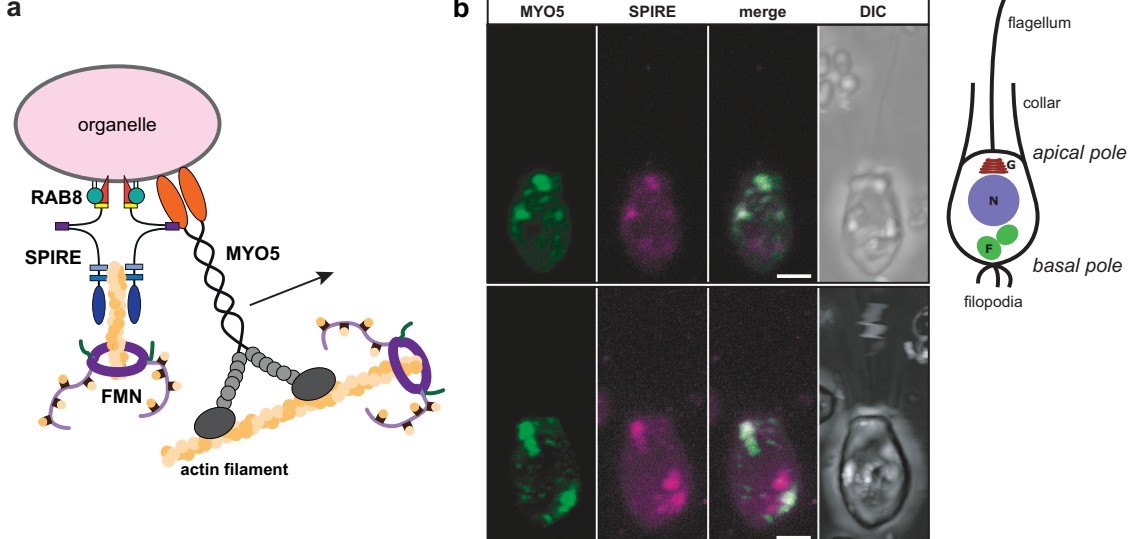

**Fig. 7 | SPIRE and myosin 5 proteins colocalise in choanoflagellate cells. a** Schematic model of the choanoflagellate RAB8/SPIRE/MYO5/FMN protein complex at the surface of organelle membranes. **b** In choanoflagellate cells MYO5 and SPIRE proteins colocalise in a vesicular pattern. Representative localisation of transiently co-expressed tagged C-terminal Sr-MYO5-cc-GTD (mNeonGreen; MYO5; green) and full-length Sr-SPIRE (mScarlet-I; SPIRE; magenta) in *S. rosetta*

cells was analysed by fluorescence microscopy. Each fluorescence channel, the overlay (merge) and differential interference contrast (DIC) microscopy images of individual cells are shown. Scale bars represent 2 μm. Eight cells were imaged. A schematic representation of the cell is shown aside indicating the intracellular organisation including Golgi system (*G*), nucleus (*N*) and food vacuoles (*F*).

## Methods

### Identification and annotation of *SPIRE* genes

The *SPIRE* genes were identified and annotated using the same approach used to annotates thousands of tubulins, myosins and dynein heavy chains, described in detail there[60,93,94]. Shortly, based on the protein sequences of SPIRE in human and *Drosophila melanogaster*, further homologs were identified in TBLASTN[52] searches in the available genome and transcriptome assemblies. Hits in transcriptome assemblies were translated in the respective reading frame. Hits in genome assemblies were used to identify gene regions, and the corresponding genomic regions were submitted to AUGUSTUS[95] to obtain gene predictions. These gene predictions were then manually corrected and refined. A total of 321 SPIRE genes were assembled. All sequence-related data (domain predictions, gene structures, sequences) and references to genome sequencing centres are available at CyMoBase (www.cymobase.org)[53]. To ensure that we did not miss any sequences beyond the detection limit of TBLASTN[52], we performed searches in the SwissProt database[96] using HMMER[97] and HHBlits[98].

### Generating the multiple sequence alignment

SPIRE proteins contain multiple short sequence motifs such as WH2, GTBM and SB motifs[62], which are connected by highly divergent linker sequences. Therefore, generating multiple sequence alignments with software such as Muscle[99], MAFFT[100] or ClustalW[101] results in highly fragmented alignments, with the four WH2 domains often not aligned in the correct order and GTBM motifs not recognised at all. Therefore, we generated a preliminary alignment with ClustalW[101], which we extensively refined manually. Based on this alignment, we further corrected the gene predictions by removing wrongly predicted sequence regions and filling gaps. If SPIRE sequence gaps remained due to gaps in genome assembly, we maintained the integrity of exons 5' and 3' of the gaps. The SPIRE sequence alignment is available at CyMoBase (www.cymobase.org)[53]. Additional multiple protein sequence alignments were performed using the Clustal Omega Multiple Sequence Alignment tool (EMBL-EBI, Hinxton, UK)[102]. For better visualisation the alignments were further processed in Jalview (v2.11.3.0)[103]. The following protein sequences were used for alignments of

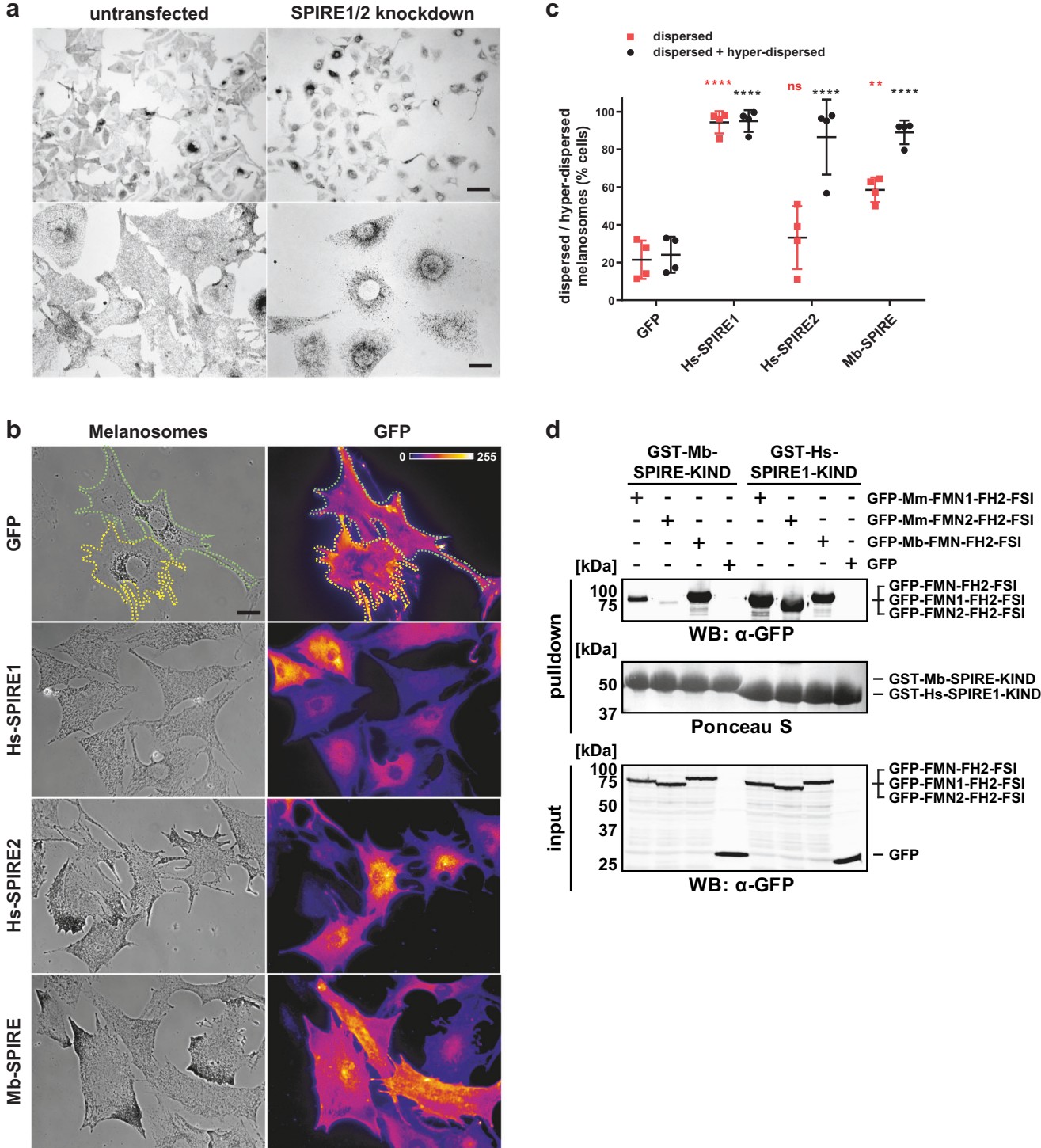

**Computing and visualising phylogenetic trees**

partial SPIRE KIND domains, RAB8 proteins, FYVE_2 domains and actin proteins: *Homo sapiens* (Hs) SPIRE1 (NP_064533.3), *Mus musculus* (Mm) SPIRE1 (XP_006526270.1), Mm-SPIRE2 (CAD30509.1), *Monosiga brevicollis* (Mb) SPIRE, *Salpingoeca rosetta* (Sr) SPIRE, Mm-RAB8A (BAF02862.1), Mb-RAB8 (XP_001750494.1), Sr-RAB8, Hs-RIM1 (NP_055804.2), Hs-RIM2 (NP_001093587.1), Hs-Rabphilin3A (NP_001137326.1), Hs-melanophilin (MLPH, NP_077006.1), *Mus musculus* skeletal muscle actin (Acta1, NP_001258970.1) and cytoplasmic actin (Actb, NP_031419.1), rabbit (*Oryctolagus cuniculus*) skeletal muscle actin (Acta1, XP_002722940.2) and *Monosiga brevicollis* actin (AAK27412.1).

The alignment was processed with CD-Hit v.4.5.4[104] using a similarity threshold of 90% to produce a dataset with less redundancy. The resulting main dataset contained 217 sequences and 2447 alignment positions. To investigate branch attraction effects, additional datasets were prepared by iteratively removing sequences of species, which resulted in deviation of the SPIRE tree in comparison to the species phylogeny (rotifera, tunicates, porifera). IQ-TREE v.1.6.beta3 ModelFinder[105] was used to determine the most appropriate amino acid substitution model. ModelFinder identified the JTT + F + R8[106] as the best model under the Bayesian Information

**Fig. 8 | Expression of choanoflagellate (*Monosiga brevicollis*) Mb-SPIRE protein rescues melanosome dispersion in mouse melanocytes depleted of endogenous SPIRE1/2.** Melan-a cells (wild-type melanocytes) were depleted of endogenous SPIRE1/2 by siRNA transfection and 72 h later infected with adenoviruses expressing the indicated proteins. Cells were fixed after 24 h and imaged using bright-field and fluorescence optics to assess localisation on melanosomes (see experimental procedures). **a** Bright-field microscopic images of wild-type (untransfected) melanocytes (left panel) and of those depleted of endogenous SPIRE1/2 (SPIRE1/2 knockdown, right panel). Images in the lower panel show higher magnification. The scale bar in the upper panel represents 50 μm and the scale bar in the lower panel represents 10 μm. **b** Representative images show distribution of melanosomes (phase-contrast microscopy) and eGFP (GFP) (fluorescence microscopy) in fields of cells expressing respective protein. Scale bar represents 10 μm. Yellow and green dotted lines in the images showing eGFP-expressing cells indicate cell edges. **c** Scatter plot showing the percentage of melanocytes in each population in which melanosome distribution is classified as dispersed and hyper-dispersed. Each data point indicates the percentage of cells in each class in each of 4 independent experiments and represents the average percentage of cells with the indicated phenotype. One hundred cells were scored for each expressed protein in each experiment. ****, ** and ns indicate significant differences $p = < 0.0001$, $<0.01$ and $>0.05$ compared with eGFP-expressing cells as determined by one-way ANOVA. Coloured asterisks indicate the class of data tested. No significant differences were seen between other datasets and GFP or Hs-SPIRE1. Bars indicate the mean and 25th and 75th percentile of data. **d** Cross-species interactions of *Monosiga brevicollis* and mammalian SPIRE and FMN proteins were analysed by GST-pulldown experiments. The SPIRE KIND domains were fused to GST and purified from bacteria. The FMN-FH2-FSI proteins were transiently expressed as AcGFP1 (GFP) fusion proteins in HEK 293 cells. Western blots (WB) of the loaded and pulled proteins and a Ponceau S stain of the GST-KIND proteins are shown.

Criterion (BIC). Phylogenetic trees were constructed using the Maximum likelihood (ML) method. ML analyses with estimated proportion of invariant sites and bootstrapping (1000 replicates) were performed using IQ-TREE[105] and FastTree[107]. The trees generated by IQ-TREE[105] showed identical topology of almost all branches and slightly varying bootstrap values (1000 replicates) at some of the less supported nodes. FastTree v.2.1.10 SSE3[107] was used with settings for increased accuracy (-spr 4 -mlacc 2 -slownni) as described in the documentation. Phylogenetic trees were visualised with FigTree (http://tree.bio.ed.ac.uk/software/figtree/).

### WebLogo sequence alignment
WebLogos[108,109] were generated online using the WebLogo server (https://weblogo.berkeley.edu) from the corresponding parts of selected vertebrate FMN1/2 sequences to represent conserved amino acids within the FMN-FSI motif, and from selected vertebrate SPIRE1/2 and choanoflagellate SPIRE sequences to represent conserved amino acids within the SPIRE-GTBM.

### Protein structure predictions
Protein structure predictions of Mb-SPIRE-KIND (amino acids 9–183) and Mb-actin were performed using ESMFold[58]. Blind (without use of PDB templates) individual structure predictions of Mm-SPIRE1-FYVE_2 (amino acids 521–730), Mb-SPIRE-FYVE_2 (amino acids 381–521) and Mb-RAB8 followed by modelling the interaction of Mb-SPIRE-FYVE_2:Mb-RAB8 were achieved by ColabFold/AlphaFold2 (individual stuctures) and ColabFold/AlphaFold-multimer (complex structure) (ColabFold v1.5.3[66–68]). This was possible by utilizing FASTA amino acid sequences of the respective protein/peptide as the only inputs for the predictions. AlphaFold-multimer produces five models and energy minimizes the best model which is presented by colouring according to a pLDDT residue-confidence score (0–100) with blue (>90, very high prediction confidence) and light blue (80, high), green (70, moderate), yellow (60, low) and red (<50, very low). The Mb-RAB8 model was superimposed on human RAB8 (PDB-ID: 4LHW[110]) using ChimeraX 4.1[111] as a benchmark for the structure prediction method used. The top-ranked model of Mm-SPIRE1-FYVE_2 exhibits a pLDDT score of 67.6 and pTM score of 0.55, demonstrating nearly good confidence in the prediction for most of the SPIRE-box (SHD1) and the FYVE_2 domain with the exception of the specific insert region, as well as demonstrating good prediction confidence for the short sequence at the beginning of helix 2 (SHD2). Similarly, the top-ranked Mb-SPIRE-FYVE_2 model with a pLDDT score of 75 and a pTM score of 0.66 exhibits good confidence in the prediction for most of the structure. The top-ranked model of Mb-RAB8 with a pLDDT score of 86 and pTM score of 0.81 was predicted with good confidence for most of the structure except the very short sequence at the start of the N-terminus, the C-terminus of the protein and the undefined regions located in the core of the structure. The top-ranked model of Mb-RAB8 in complex with Mb-SPIRE-FYVE_2 shows a pLDDT score of 80.9, a pTM score of 0.77, and an ipTM score of 0.77 indicating good confidence in the prediction for both structures and reveals a contact interface consisting mainly of the SPIRE-box and switch 1

and switch 2 regions of RAB8. The structure of Mb-MYO5-GTD in complex with the Mb-SPIRE-GTBM was predicted by ColabFold/AlphaFold-multimer (ColabFold v2.3.5[66–68]). The top ranked solution with pLDDT = 91.5, pTM = 0.91, and ipTM = 0.876 was predicted with a high confidence for the MYO5-GTD and the major part of the SPIRE-GTBM. The middle part of the GTBM is predicted to be flexible and forming a bulge not participating to the MYO5 binding. Predicted protein structures were further processed in the PyMOL Molecular Graphics System (v2.5.4; Schrödinger, LLC). Structural alignments of Mb-actin and actin: SPIRE-WH2 (PDB-ID: 3MN5[57]), Mb-SPIRE-KIND and SPIRE1-KIND:FMN2-FSI (PDB-ID: 2YLE[17]), Mb-SPIRE-FYVE_2 and Mm-SPIRE1-FYVE_2, and Mb-MYO5-GTD:Mb-SPIRE-GTBM and Hs-MYO5A-GTD:Hs-SPIRE2-GTBM (PDB-ID: 5JCY[28]) were performed in PyMOL.

### Gene synthesis of choanoflagellate cDNAs
Full-length cDNA coding regions of *Mb-SPIRE* and *Mb-RAB8* and a coding region of the *Mb-MYO5* globular tail domain and *Mb-FMN* C-terminal sequences (FH2-FSI) were codon-optimised for expression in *E. coli* bacterial cells and generated by gene synthesis as cDNAs (Eurofins Genomics Germany, Ebersberg, Germany). Synthesised cDNA sequences were subsequently used as templates for further subcloning of respective cDNA fragments to allow for bacterial and eukaryotic protein expression.

### Expression vector cloning
Expression vectors were generated by standard cloning techniques using AccuPrime Pfx DNA polymerase (Thermo Fisher, Waltham, MA, USA), restriction endonucleases and T4 DNA ligase (both New England Biolabs, NEB; Frankfurt am Main, Germany). Point mutations were generated using the In-Fusion HD cloning kit (TakaraBio/Clontech, Saint-Germain-en-Laye, France). For generation of the pGEX-6P1-SNAP-Mm-SPIRE1-KIND-WH2-ΔB,C vector an assembly PCR was performed using primers complementary to sequences prior and after the coding regions for WH2-B and WH2-C to amplify a cDNA fragment missing the Mm-SPIRE1-WH2-B/-C region. The resulting fragment is devoid of the sequence corresponding to the amino acid sequence ranging from Pro285 to Thr349 of the mouse SPIRE1 protein (XP_006526270.1). The amplified fragment was inserted into pGEX-6P1-SNAP by 5'-BamHI and 3'-EcoRI restriction sites. DNA sequencing was carried out by LGC Genomics (Berlin, Germany). pKanCMV-mClover3-mRuby3 was a gift from Michael Lin (Addgene plasmid # 74252; http://n2t.net/addgene:74252; RRID:Addgene_74252)[112] and was used as template for subcloning of mRuby3 cDNA. Supplementary Table 1 shows details on cloning of vectors used in this study.

### Protein purification
Recombinant GST, GST-Mb-MYO5-GTD, GST-Sr-MYO5-GTD, GST-Mb-FMN-eFSI, GST-Mb-SPIRE-KIND, GST-Hs-SPIRE1-KIND, GST-Mb-RAB8-Q67L and GST-Mb-RAB8-T22N proteins were expressed in *E. coli* Rosetta bacterial cells (Merck Millipore, Novagen, Darmstadt, Germany). Bacteria were cultured in LB medium (100 mg/l ampicillin, 34 mg/l

chloramphenicol) at 37 °C until an $OD_{600nm}$ of 0.6 - 0.8. Protein expression was induced by 0.2 mM Isopropyl-β-D-thiogalactopyranoside (IPTG; Sigma-Aldrich, Taufkirchen, Germany) and continued at 20 °C for 18 - 20 h. Bacteria were harvested and lysed by ultra-sonication in lysis buffer (1 x PBS; 2 mM β-mercaptoethanol; 2 mM $MgCl_2$; 0.05% (v/v) Tween-20; 5% (v/v) glycerol; 1 mM PMSF; protease inhibitor cocktail). Soluble proteins were purified using glutathione-Sepharose 4B beads (GE Healthcare Life Sciences, Freiburg, Germany) and size exclusion chromatography (High Load 16/60 Superdex 200; GE Healthcare Life Sciences). Proteins were concentrated by ultrafiltration using Amicon Ultra centrifugal filters (Merck Millipore) with respective molecular weight cut offs. The final protein purity was estimated by sodium dodecyl sulfate-polyacrylamide gel electrophoresis (SDS-PAGE) and Coomassie staining. N-terminal protein fragments for vertebrate (*Mus musculus*) and choanoflagellate (*Monosiga brevicollis*) SPIRE proteins including KIND and WH2 sequences used in actin assembly, disassembly and TIRF assays were expressed as GST- and SNAP-tag fusions by pGEX-6P1-SNAP-tag vectors under conditions as described above followed by cleavage of the GST-tag. Here, 10 mg of purified GST-SNAP-tagged protein was mixed with 85 μl (170 units) PreScission Protease (Cytiva, Munich, Germany) in cleavage buffer (50 mM Tris-HCl, pH 7.0; 150 mM NaCl; 1 mM EDTA; 1 mM DTT). Cleavage was done in a total volume of 22 ml, supplemented with 0.01% (v/v) Triton-X100 over night at 4 °C on a rotating wheel. After protease incubation 1 ml glutathione-Sepharose 4B beads were added to the solution and incubated for 2 h at 4 °C to remove GST and uncleaved GST-fusions, respectively. Beads were pelleted and the supernatant was concentrated by ultrafiltration using Amicon Ultra centrifugal filters (Merck Millipore) with 50 kDa cut off. For all proteins small aliquots were flash frozen in liquid nitrogen and stored at −80 °C for further use. His-tagged *Dictyostelium discoideum* heterodimeric capping protein Cap32/34 (CP)[113] was purified by standard procedures using Ni-NTA resin (Qiagen, Hilden, Germany) followed by size exclusion chromatography using a preparative 26/60 Superdex S200 column (GE Healthcare Life Sciences). Actin was extracted from rabbit skeletal muscle according to Spudich and Watt[114], and was labeled on Cys374 with ATTO488.

### Pyrene-actin assays
The polymerization of 2 μM rabbit skeletal muscle G-actin (5% pyrene labelled) alone or in the presence of different SPIRE proteins as indicated was initiated with 1 x KMEI (50 mM KCl; 1 mM $MgCl_2$; 1 mM EGTA; 10 mM imidazole, pH 7.4) and was monitored in 96-well plates using a Synergy 4 fluorescence microplate reader (BioTek/Agilent, Waldbronn, Germany). For assaying dilution-induced depolymerization experiments, G-actin (50% pyrene labelled) was first polymerised in 1 x KMEI over night at 4 °C. Subsequently, the F-actin (0.5 μM) was diluted to the critical concentration of 0.1 μM in the absence or presence of SPIRE proteins into the wells using the automated dispenser of the Synergy 4 plate reader and analysed as above.

### TIRF microscopy
For the in vitro actin assembly TIRFM experiments, the SNAP-tagged SPIRE proteins were prediluted in 2 x TIRF buffer (40 mM imidazole, pH 7.4; 100 mM KCl; 2 mM $MgCl_2$; 2 mM EGTA; 40 mM β-mercaptoethanol; 0.4 mM ATP; 30 mM glucose; 0.5 mg/ml methylcellulose (4000 cP); 40 μg/ml catalase; 200 μg/ml glucose oxidase). The TIRF assays were performed in 1 x TIRF buffer and initiated by addition of ATP-G-actin (1 μM final concentration, 20% Atto488 labeled) and by flushing of the mixtures into mPEG-silane (Mr 2000) (Laysan Bio) pre-coated flow chambers. For in vitro actin disassembly TIRF experiments, 10 μM G-actin (20% Atto488 labeled) was first polymerised in 1 x KMEI over night at 4° C. Subsequently, the flow chambers were pre-incubated with 0.5% cold fish gelatin (Sigma) in 1 x KMEI for 10 min and then rinsed with 1 x KMEI containing 10 mg/ml BSA. The TIRF assays were initiated by flushing a mixture of ADP-F-actin (0.01 μM final concentration) in the absence or presence of Mm-SPIRE or Mm-SPIRE1-KW-ΔB,C proteins into the flow chambers. Images were

captured with a Nikon Eclipse TI-E inverted microscope equipped with a TIRF Apo 100 x objective at 3 s intervals with exposure times of 60 ms by Ixon3 897 EMCCD cameras (Oxford Instruments/Andor, Belfast, UK) for at least 10 min. The pixel size corresponded to 0.159 μm. The elongation rates of filaments were determined by manual tracking of growing barbed-ends using Fiji software[115]. At least 10 filaments were measured from three independent movies per condition. Nucleation efficiencies were determined by counting and averaging the number of nucleated actin filaments in a 50 μm x 50 μm area after 90 s in three independent experiments.

### Cell culture
HEK 293 and HeLa cells (both from ATCC, Manassas, Virginia, USA) were cultured in Dulbecco's Modified Eagle's Medium (DMEM; Thermo Fisher) supplemented with 10% (v/v) fetal calf serum (FCSIII; GE Healthcare Life Sciences, HyClone), 2 mM L-glutamine (Thermo Fisher), penicillin (100 units/ml; Thermo Fisher) and streptomycin (100 μg/ml; Thermo Fisher) at 37 °C, 5% $CO_2$, 95% humidity and were passaged regularly at 80% confluency. Transfections with plasmid DNA were performed using Lipofectamine2000 reagent (Thermo Fisher) according to manufacturer's recommendation. Cultures of immortal melan-a melanocytes (available from the Wellcome Trust Functional Genomics Cell Bank at St George's University of London UK SW17 0RE) were maintained in accord with the recommendation of the supplier (https://www.sgul.ac.uk/about/our-institutes/neuroscience-and-cell-biology-research-institute/genomics-cell-bank).

### GST-pulldown protein interaction assays
For GST-pulldowns HEK 293 cells were transfected with expression vectors encoding AcGFP1-tagged Mb-SPIRE and Sr-SPIRE full-length proteins and respective deletion mutants, and C-terminal fragments of Mb-FMN, Mm-FMN1 and Mm-FMN2, respectively. AcGFP1 alone was used as a control. 48 h post transfection, cells were lysed in lysis buffer (25 mM Tris-HCl, pH 7.4; 150 mM NaCl; 5 mM $MgCl_2$; 10% (v/v) glycerol; 0.1% (v/v) Nonidet P-40; 1 mM PMSF; protease inhibitor cocktail) and centrifuged at 20,000 x g, 4 °C, 20 min to remove insoluble debris. 65 μg GST-Mb-MYO5-GTD, 65 μg GST-Sr-MYO5-GTD, 32 μg GST-Mb-FMN-eFSI, 47 μg GST-Mb-RAB8-Q67L/T22N, 50 μg GST-Mb-SPIRE-KIND and 50 μg GST-Hs-SPIRE1-KIND, respectively, and 20–25 μg GST protein as control were coupled to glutathione-Sepharose 4B beads (1:1 suspension) for 1 h, 4 °C on a rotating wheel. Beads were washed twice with pulldown buffer (25 mM Tris-HCl, pH 7.4; 150 mM NaCl; 5 mM $MgCl_2$; 10% (v/v) glycerol; 0.1% (v/v) Nonidet P-40) and subsequently incubated with the cell lysates for 2 h at 4 °C on a rotating wheel. Beads were washed four times with pulldown buffer and bound proteins were eluted with 1 x Laemmli buffer, denatured at 95 °C for 10 min and then analysed by immunoblotting.

### Immunoblotting
Proteins were separated by SDS-PAGE and analysed by Western blotting using anti-GFP primary antibody (Living Colors Full-length GFP Polyclonal Antibody, rabbit polyclonal, diluted 1:1000 in 5% milk powder solution; TakaraBio/Clontech) and horseradish peroxidase linked anti-rabbit IgG secondary antibody (from donkey, diluted 1:5000; GE Healthcare Life Sciences). Protein bands were visualised by chemiluminescence (Luminata Forte Western HRP substrate; Merck Millipore) and recorded with an Image Quant LAS4000 system (GE Healthcare Life Sciences). Recorded images were processed in Adobe Photoshop and assembled in Adobe Illustrator.

### Cell fixation for fluorescence microscopy
HeLa cells were seeded on microscope cover glasses and transfected to transiently express fluorescent proteins or fluorescent protein-tagged Mb-SPIRE, Mb-SPIRE-ΔKW, Mb-FMN-FH2-FSI, Mb-RAB8 and Mb-MYO5-GTD proteins. Cells were fixed with paraformaldehyde (3.7% in 1 x PBS) for 20 min at 4 °C and subsequently mounted on microscope slides with Mowiol, dried at room temperature in the dark and stored at 4 °C.

## Fluorescence microscopy

Fixed cells were analysed with a Leica AF6000LX fluorescence microscope (Leica, Wetzlar, Germany), equipped with a Leica HCX PL APO 63x/1.3 GLYC objective and a Leica DFC7000GT CCD camera (12-bit, pixel size: 4.54 × 4.54 μm, 1920 × 1440 pixels, 2 × 2 binning mode). 3D stacks were imaged and processed with the Leica deconvolution software module. Images were recorded using the Leica LASX software and further processed with Adobe Photoshop and subsequently assembled with Adobe Illustrator.

## Cloning of *Salpingoeca rosetta* expression vectors

The choanoflagellate expression vector NK644 was modified for fluorescence microscopy studies in *S. rosetta*. NK644 was a gift from Nicole King (Addgene plasmid # 109096; http://n2t.net/addgene:109096; RRID:Addgene_109096)[72]. The vector was first digested with *Bam*HI and *Eco*RI restriction endonucleases. Sticky ends were filled with Klenow polymerase (NEB) to destroy an existing multiple cloning site downstream of the non-coding *Salpingoeca rosetta* non-muscle actin sequences. mNeonGreen[116] and mScarlet-I[117] cDNA sequences were codon-optimised for expression in *S. rosetta* and subsequently synthesised (Eurofins Genomics Germany) including a C-terminal extension containing a newly designed multiple cloning site for insertion of downstream gene sequences. The existing mCherry-geranyl-geranyl sequences from the modified NK644 vector were substituted by the codon-optimised mScarlet-I- and mNeonGreen-MCS sequences, respectively. *S. rosetta* MYO5-cc-GTD or SPIRE cDNA sequences were inserted subsequently into the vector backbones using the new MCS (*Eco*RI and *Bam*HI). Codon optimization of mNeonGreen and mScarlet-I cDNA sequences was done with the Gene Optimizer Server (http://genomes.urv.es/OPTIMIZER/)[118] employing the *S. rosetta* codon usage table for highly expressed intronless genes provided by Booth and colleagues[72]. 3xmNeonGreen_UtrCH was a gift from Dorus Gadella (Addgene plasmid #129607; http://n2t.net/addgene:129607; RRID:Addgene_129607)[119] and was used for subcloning of the mNeonGreen cDNA. The pmScarlet-I-C1 vector was used as template for initial subcloning of the mScarlet-I cDNA sequence and was a gift from Dorus Gadella (Addgene plasmid # 85044; http://n2t.net/addgene:85044; RRID:Addgene_85044)[117].

## *Salpingoeca rosetta* culturing

*S. rosetta was* co-cultured with a single bacterial strain *Echinicola pacifica* as food source[120,121] (ATCC). To maintain the cultures, cells were incubated at 22 °C and were passaged every 48 h at a final volume of 8000 cells/ml into 6 ml of High Nutrient Media[72] in 25 cm² TC vented flasks (Sarstedt, Nümbrecht, Germany).

## Transfection of *Salpingoeca rosetta*

(1) Cell culture—the transfection protocol was adopted from ref. 72. Two days before transfection, two 75 cm² TC vented flasks (Sarstedt) containing 30 ml High Nutrient Media were inoculated with *S. rosetta/E. pacifica culture* to a final concentration of *S. rosetta* at 8000 cells/ml per bottle and incubated at 22 °C. (2) Washing—after 36–48 h cells were shaken vigorously for 10 s and transferred to 50 ml falcon tubes, shaken again for 10 s, and centrifuged for 5 min at 2000 x *g* at 22 °C in a centrifuge with a swinging bucket rotor. The supernatant was carefully removed using a serological pipette and the pellet was resuspended in fresh 10 ml artificial seawater[73] and centrifuged for 5 min at 2000 x *g* at 22 °C. This step was repeated twice. The pellet obtained at the end of the third centrifugation step was resuspended in 200 μl artificial seawater. To measure the cell concentration, 2 μl of washed cells were suspended in 196 μl of artificial seawater containing 2 μl of 37.5% formaldehyde, vortexed, and counted on a Luna-FL automated cell counter (Logos Biosystems, Anyang, KOR). Cells were diluted to a final concentration of $5 \times 10^7$ cells/ml and distributed into 100 μl aliquots ($5 \times 10^6$ cells per aliquot). (3) Priming—aliquots were pelleted at 800 x *g* and 22 °C for 5 min. Supernatants were gently removed and resuspended in 100 μl of priming buffer (40 mM HEPES-KOH, pH 7.5; 34 mM lithium citrate; 50 mM l-cysteine; 15% (wt/vol) PEG 8000); 2.5 μM Papain (Millipore

Sigma, St. Louis, MO, USA). Cells were incubated for 30–40 min at room temperature. 10 μl of 50 mg/ml bovine serum albumin was added to quench proteolysis from the priming buffer. Finally, cells were centrifuged at 1250 x *g* and 22 °C for 5 min, the pellet was resuspended in 25 μl SF Buffer (Lonza, Basel, Switzerland) and stored on ice. (4) Nucleofection—14 μl of Lonza SF Buffer DNA mixture (2 μl 20 μg/μl pUC19; 1 μl each of two reporter plasmids at 5 μg/μl), and 2 μl of primed cells were thoroughly mixed and transferred to a well of a 16-well cuvette of the Lonza SF Cell Line 4D Nucleofection Kit (Lonza). Cells were transfected in the 4D-Nucleofector X Unit (Lonza) by applying the CM-156 pulse. (5) Recovery—immediately after nucleofection, ice-cold recovery buffer (10 mM HEPES-KOH, pH 7.5; 0.9 M sorbitol; 8% (wt/vol) PEG 8000) was added to the cells. 5 min after recovery at room temperature, cells were transferred to 6 well plates containing 2 ml of Low Nutrient Media[72] in each well. After 30 min, 10 μl of 10 mg/ml of *E. pacifica* (prepared by resuspending a frozen 10 mg pellet of *E. pacifica* in artificial seawater) was added to each well. Cells were incubated at 22 °C and 60% relative humidity for 48 h.

## *Salpingoeca rosetta* fluorescence microscopy

35 mm glass-bottom dishes (MatTek Dishes, Ashland MA, USA) were treated with 100 μl of Poly-L-lysine for 10 min and washed three times with 150 μl of artificial seawater. Transfected cells were centrifuged at 3600 x g at 4 °C for 10 min in a swinging bucket rotor. The supernatant was removed, and the pellet was resuspended with 100 μl of 100 mM LiCl in artificial seawater to reduce flagellar beating. 50 μl of resuspended cells were added to Poly-L-lysine coated dishes and incubated for 10 min for cells to settle. Drop by drop, 50 μl of imaging solution (20% (wt/vol) Percoll® (Merck Life Science AS, Oslo, Norway), 100 mM LiCl in artificial seawater) was added to the dish. The cells were imaged at 60x objective with silicone immersion oil (Olympus Scientific Solutions, Tokyo, Japan) using FLUOVIEW FV3000 Confocal Laser Scanning Microscope (Olympus Scientific Solutions).

## Melanosome dispersion assay

melan-a cells (wild-type melanocytes) were depleted of endogenous SPIRE1/2 by siRNA transfection and 72 h later infected with adenoviruses expressing siRNA resistant human and choanoflagellate SPIRE proteins. Cells were fixed 24 h later, processed for immunofluorescence and imaged using bright-field, phase-contrast and fluorescence optics to observe melanosome and protein distribution/expression. Melanosome distribution was scored categorically as 'dispersed' if melanosomes were evenly distributed throughout the cytoplasm and 'hyper-dispersed' if melanosomes were cleared from the cell centre and enriched in the periphery. siRNA transfection of melanocytes was performed using Oligofectamine (Invitrogen, UK)[122]. The sequence of siRNA oligonucleotides was GGACGACAUUCGGUGCAAA for mouse SPIRE1 and CAAAGAACACUGCACGAGA for mouse SPIRE2. All siRNA oligonucleotides were from Sigma Genosys UK. Adenovirus expression vectors were generated using the Invitrogen ViraPower™ Adenoviral Expression System[123].

## Statistics and reproducibility

Quantitative actin assembly TIRF experiments were performed at least in triplicates to avoid any potential environmental bias or unintentional error. All data sets were tested for normality by the Shapiro-Wilk test. Statistical differences between normally distributed datasets of more than two groups were examined by one-way ANOVA and Tukey Multiple Comparison test. In case of not normally distributed data, the nonparametric Kruskal-Wallis test was used. All values are shown as means ± SEM. Statistical analysis for the quantification of melanosome distribution in melanocytes was performed using one-way ANOVA and Tukey's multiple comparisons posttest facility within the Graphpad Prism 6 software. For the GST-pulldown assays two to three independent experiments were performed. For fluorescence microscopy colocalization analysis six cells from two independent experiments were imaged. For live cell fluorescence microscopy of *Salpingoeca rosetta* choanoflagellates 16 cells showing fluorescence were imaged from three independent experiments. Here, eight cells showed SPIRE/MYO5 colocalisation, a reasonable level of

fluorescence-labeled protein expression and normal morphology. These cells were included in the analysis. To study melanosome distributions in melanocytes four independent experiments were performed. In each of these experiments 100 cells per condition were scored. Each experimental repeat is defined as including all methodical steps required for the experimental data collection. For the GST-pulldown experiments this included HEK 293 cell seeding, transfection, preparation of cell lysates, GST-pulldown and sample preparation, SDS-PAGE and protein transfer, Western blotting with antibody treatment and band detection via chemiluminescence. For fluorescence microscopy studies this included HeLa cell seeding, transfection, fixation and mounting, fluorescence microscopy and image recording and image processing. For melanosome dispersion assays this included melan-a cell seeding, siRNA transfection, adenovirus infection, fixation and image recording.

## Data availability

The data underlying this article are available in the article and in its online supplementary material. Uncropped and unedited Western blot images, Ponceau S stainings and Coomassie gel stainings are presented as supplementary material (Supplementary Figs 6–12). Numerical source data for actin assembly and disassembly assays are shown in Supplementary Data 1, for TIRF microscopy related bar diagrams in Supplementary Data 2, and for melanosome dispersion related scatter blots in Supplementary Data 3. All sequence-related data (domain predictions, gene structures, sequences) and references to genome sequencing centres are available at CyMoBase (www.cymobase.org)[53].

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

## Acknowledgements

M.K. would like to thank Prof. Christian Griesinger for his continuous generous support. We thank Felix Straub for his help with the art work. This project has been funded by grants SPP 1464: KO 2251/13-1 (to M.K.) and SPP 1464: K.E. 447/10 −2 and K.E. 447/18-1, 21-1 (to E.K.) and 330/12-3 (to J.F.) of the Deutsche Forschungsgemeinschaft (D.F.G.). AR and PB were supported by the Michael Sars Centre core budget. OP was supported by Curie Institute, CNRS and ANR-20-CE18-0016-02. The work of AA was supported by a PhD studentship from the Princess Nourah Bint Abdulrahman University, Kingdom of Saudi Arabia. NA was supported by a PhD studentship from Taibah University, Medina, Kingdom of Saudi Arabia. AH and DAB were supported by the Leverhulme Trust RPG-2022-158 Project Grant, Wellcome Trust Grant 204843/Z/16/Z and Medical Research Council New Investigator Award G1100063.

## Author contributions

M.K. and E.K. have initiated the project. M.K. and K.H. have performed the bioinformatic analysis of SPIRE protein evolution. T.W., A.R., T.K., N.A., J.K., D.A.B., and A.S.W. have carried out the molecular cell biology and

biochemistry experiments. T.W., A.A., and O.P. carried out protein structural research. M.K., T.W., A.R., T.K., O.P., A.H., P.B., J.F. and E.K. have designed the research and contributed to the writing of the manuscript. All authors read and approved the final manuscript.

## Funding

## Competing interests
The authors declare no competing interests.
