## [Peer Review File · Communications Biology]

Reviewers' comments:

Reviewer #1 (Remarks to the Author):

This study addresses the question of the evolution of the SPIRE, FMN and Myo5 set of protein interaction among members of the holozoan clade, and discusses its relevance in vesicle trafficking. To date, SPIRE proteins have not been described outside the animal kingdom and it is not known whether exocytic actomyosin transport mechanisms exist in other eukaryotes. The paper identifies the homologs of SPIRE, FMN and MYO5 in one choanoflagellate (the closest unicellular relatives of animals) species, and in a more distantly related unicellular protists called ichthyosporeans. This allowed them to conclude that the organelle-associated cooperative function of SPIRE actin nucleators and MYO5 motor proteins precedes animal evolution.

The study is well conducted and the paper clearly progresses through the different protein-protein interactions, and co-localization in cells.

I have only a few major points I would like the authors to address and discuss, which may be useful for the readers as well:

- When introducing the different different proteins domains of SPIRE, the authors could mention if these domains are known to be structured or disordered regions.
- Related to the previous point: one can regret that, in the era of computer-assisted protein folding (AlphaFold, EMSFold), there was no attempt to compare 3D structure of domains and known binding sites between the different homologs.
- The conclusion from actin-based assays is: "As opposed to Mm-SPIRE1-KW, Mb-SPIRE-KW is a weak nucleator." The authors should clearly indicate that this has been performed using rabbit skeletal actin, and not organism-specific actin proteins. Still, I agree that the number of WH2 domains is already strongly indicating whether a SPIRE protein will be a 'good' or 'weaker' actin nucleator.
- Related to the previous point: how identical are the actins isoforms in between these organisms. It would be interesting to discuss and document this, to better discuss the importance of SPIRE on actin dynamics.
- Actin disassembly assay: this could also be performed in TIRF microscopy to infer the molecular mechanism of accelerated disassembly (filament fragmentation, faster barbed or pointed end depolymerization) ?
- figures: please show individual experimental repeats as data points, not only the average and SEM values in graphs.
- I do not understand the conclusion "The protein complex may have contributed to the evolution of animals and their sophisticated networks of polarised and communicating cells by providing an extended toolkit of actin and myosin functions in intracellular transport processes." Since this machinery is also present in other holozoans, even though the multiplicity of SPIRE and RAB orthologs would expand the toolkit size. Can the authors explain more clearly how they would see the SPIRE/FMN/MYO as a driver of evolution limited to animals only ?

Minor point:

There are a number of writing issues along the text, like (p7) "Actin nucleation was found to be required an array of at least two WH2 domains, ...", (p8) "Mouse SNAP-SPIRE1- KW on the other hand induced actin filament assembly much more efficient already at lower concentration, ...", (p15) "which serve as tracks for myosin motor protein mediating centrifugal dispersion of the vesicles ..."

Reviewer #2 (Remarks to the Author):

In this manuscript, Kollmar et al. investigate the evolutionary origins of SPIRE actin nucleators. They uncover SPIRE homologs in two unicellular organisms, choanoflagellates and ichthyosporeans. The choanoflagellate SPIRE is conserved enough to rescue the function of mammalian SPIRE in a knockdown system. Choanoflagellate SPIRE nucleates actin polymerization and maintains interaction partners essential for SPIRE function in mammals, including formin and class-5 myosin. Additionally, SPIRE co-localizes with RAB8, suggesting it forms a protein complex similar to that seen with SPIRE1 and RAB27A in mammals. Together, these findings show the SPIRE/myosin/formin protein complex originated in unicellular ancestors and has a conserved function in exocytic membrane transport.

This work provides important insight into the evolutionary origins of a core member of membrane trafficking by taking an interdisciplinary approach. The authors leveraged multiple *in vivo* and *in vitro* systems to demonstrate that SPIRE function is more ancient than previously appreciated. These findings will interest both those who study membrane transport and evolutionary biologists. However, the paper is not written for a broad readership and could significantly improve in clarity and organization. Additionally, some data detract from the core findings. Our specific comments are as follows:

Major comments:

1. We found that the organization of the manuscript, particularly the results sections pertaining to Figures 3 and 4, needs to be easier to follow, especially for someone outside the field. It seems the data in Figures 3 and 4 could be better organized by separating the findings for each actin regulator by figure (rather than describing all interactions in one figure). Details also appeared to be repetitive, appearing in the introduction and the results. Given the length of the intro, which seems more like a review, information could be introduced where it is most pertinent in the results sections.
2. In the evolutionary analysis, it should be made clear in the text that the protein sequences are classified as SPIRE1-3 based on percent identity and phylogenetic analysis. It was not shown that the genes obtained by tBLASTn are syntenic; therefore, it is unknown if all 'SPIRE1' proteins (for example) are truly orthologous. Furthermore, domains are often described as 'SPIRE-like' or distinguished by color. However, it would be helpful to know the protein identity relative to whichever species the authors used as their reference.
3. In Figure 5, the authors report GFP-tagged SPIREs rescue melanosome dispersion. However, following siRNA knockdown, cells with melanosome dispersion do not always appear GFP fluorescent. For example, the image of Mb-SPIRE melanosomes shows a cell in the upper left corner with hyperdispersion, yet it does not appear to express GFP-tagged Mb-SPIRE. The authors need to provide data, such as a western blot, to show how successful the knockdown was, which may clarify the discrepancy in the image.
4. The analysis of human SPIREs and their Rab specificity detracted from the main focus on unicellular SPIRE. The fact that Choano SPIRE failed to interact with human Rab3 and Rab27, which evolutionarily arose after Rab8, was not surprising. Figure 6 could contribute to a different paper; alternatively, the framing of why it belongs in this story needs to be clarified.

Minor comments:

1. The authors note that the observed differences in Figure 2's *in vitro* work may be due to the use of rabbit actin. The authors should list the percent identities for Mb actin, mouse actin, and rabbit actin for readers to assess how likely the *in vitro* differences are due to species-specific actin differences.
2. On page 12, the description of Myo5 motility in choanos seems to be an aside and is not pertinent to the story of SPIRE.
3. Please define acronyms when used the first time (for example, 'WH2' on page 3).
4. Some comments made throughout the results section are better placed in the discussion.
5. The schematic for Amoebozoa in Figure 1B and C should match for clarity.

6. Figure 1: It may be better to note the WH2 domains as A-D in the text, rather than by different shades of blue.
7. In Figure 3A, why is mouse Formin and SPIRE used, yet Human Myo5a is shown?
8. The model in Figure 3F was never referenced in the text, yet it was useful. Its description may help when clarifying the story's findings.
9. The model in Figure 4D needs better labeling, such as the 'apical end.'

Reviewer #3 (Remarks to the Author):

Review to

"Actomyosin organelle functions of SPIRE actin nucleators precede animal evolution"

Cytoskeleton-based cargo transport exists in widely diverse organisms. Understanding the evolution of the proteins involved can improve the understanding of the transport process. Although the role of SPIRE protein in coordinating actin filament assembly and myosin motor-dependent force generation is well-studied in specific metazoans (mammals, flies), the evolutionary origin of the SPIRE remains unclear. Here, Kollmar et al. studied the origin of SPIRE by phylogenetic and biochemical approaches, revealing that SPIRE function originated in the last common ancestor of holozoans, which includes animals and their closest single-celled relatives, choanoflagellates. The data shown support that SPIRE from choanoflagellates is able to interact with formin, MYO5, and Rab GTPase, which also shares the similar function as mouse SPIRE protein. Overall, the data shown by the authors support their conclusions and clarifies the origin and function of the SPIRE protein.

This manuscript provides an interesting analysis of SPIRE across a wide evolutionary range, and as such is a very useful contribution to our knowledge of these proteins. There are several issues that require resolution, however.

Comments:

1. The authors mentioned that SPIRE is not universally conserved among Holozoa, e.g., quote 'SPIRE has been lost in the unicellular filasterean'. The authors may want to clarify for this point: is the SPIRE sequence clearly not present in the filasterean genome or it is not accessible in the database?
2. The illustration in Fig. 1b is confusing for the SPIRE part, since there are differences between SPIRE isoforms, especially for the WH2 domain, I would not recommend drawing exactly two WH2 diamonds there. Instead, the author may use a box to indicate all WH2s to avoid confusion.
3. Fig. 2 c,d. These actin depolymerization assays need more controls:
 - a. Phalloidin: as a depolymerization-blocking control
 - b. LatA or LatB: This would show the effect of a compound that only sequesters.
4. In Fig. 3d (pull-down assay), there are two issues:
 - a. the pull-down assay shows a relatively weak binding, while the authors state the FMN-FSI protein strongly interacted with Mb-SPIRE. More conservative wording would be better
 - b. one reason for weaker binding could be due to Mb-SPIRE autoinhibition by interacting KIND domain with FYVE2 domain. The authors may want to pull down Mb-FMN and RAB8 with SPIRE-KW or SPIRE-FYVE2.
5. For Fig. 3e colocalization assay, the author should use SPIRE- Δ KW and FMN-FH2-FSI as a good

negative control.

6. It is not clear how the data of Fig. 4c alone supports the conclusion that the mammalian and Choanoflagellate GTBMs are distantly related. More explanation would be required.

7. For the rescue experiment Fig. 5a, there are two concerns:

a. The Melanosomes in WT melanocytes must also be shown. Showing melanosomes in the rescue group is not sufficient;

b. It is interesting that in Hs-SPIRE2 rescue group, on the left hand, there is one cell doesn't show any SPIRE/GTP signal, but shows a hyper-dispersed phenotype. Can the authors explain why?

8. Since the authors compared the binding of SPIREs with formin, can the authors briefly describe the difference between SPIRE1-3 in the introduction and analyze which SPIRE isoform is more close to the MbSPIRE?

COMMSBIO-23-3133A - Response Letter

Dear reviewers,

We would like to thank you for the fair reviewing process and the very helpful comments. We have addressed the comments and please find a point to point response below.

In summary, we have put a lot of work into the revision including new actin biochemistry experiments, protein interaction studies and protein structure predictions. Two additional authors were included, Asmahan Alghamdi and Olena Pylypenko, who both contributed to protein structure predictions. Felix Straub is no longer an author of the paper because we have removed the mammalian SPIRE1/2 - RAB interaction studies, as suggested by reviewer #2.

We have put much effort in the rewriting of the manuscript. To improve the English the manuscript has been edited by a native speaker. Sections of the manuscript, which underwent major revision are marked in red font color.

Reviewers' comments:

Reviewer #1 (Remarks to the Author):

This study addresses the question of the evolution of the SPIRE, FMN and Myo5 set of protein interaction among members of the holozoan clade, and discusses its relevance in vesicle trafficking. To date, SPIRE proteins have not been described outside the animal kingdom and it is not known whether exocytic actomyosin transport mechanisms exist in other eukaryotes. The paper identifies the homologs of SPIRE, FMN and MYO5 in one choanoflagellate (the closest unicellular relatives of animals) species, and in a more distantly related unicellular protists called ichthyosporeans. This allowed them to conclude that the organelle-associated cooperative function of SPIRE actin nucleators and MYO5 motor proteins precedes animal evolution.

The study is well conducted and the paper clearly progresses through the different protein-protein interactions, and co-localization in cells.

I have only a few major points I would like the authors to address and discuss, which may be useful for the readers as well:

- When introducing the different protein domains of SPIRE, the authors could mention if these domains are known to be structured or disordered regions

In the revised manuscript we have included the SPIRE KIND (Fig. 4d) and WH2 domain (Supplementary Fig. 4a) structures, which were solved by X-ray crystallography, and of the SPIRE FYVE_2 structure (Fig. 5c), which was generated by artificial intelligence structure prediction. By this the reader should get a good overview on SPIRE structural domains. In addition, we have included a detailed description of the SPIRE GTBM motif (page 7): "The structurally characterised mammalian SPIRE2-GTBM is a short 26 amino acids long sequence motif in the central part of the protein, which binds to the surface of the MYO5 globular tail domain (Pylypenko et al., 2016). The GTBM consists of an array of basic, hydrophobic and acidic residues, which are not structured in the sense that they fold into secondary structural elements (Pylypenko et al., 2016)."

- Related to the previous point: one can regret that, in the era of computer-assisted protein folding (AlphaFold, EMSFold), there was no attempt to compare 3D structure of domains and known binding sites between the different homologs.

Very good consideration. We supported our protein interaction and colocalization studies with protein structure predictions and structure alignments for choanoflagellate (*Monosiga brevicollis*) SPIRE-KIND (Figure 4d) and MYO5-GTD (Figure 6b) using EMSFold, and for mouse and choanoflagellate SPIRE-FYVE_2 domains (Figure 5c) as well as the choanoflagellate SPIRE-FYVE_2/RAB8 protein complex (Figure 5d) using AlphaFold/ColabFold.

- The conclusion from actin-based assays is: "As opposed to Mm-SPIRE1-KW, Mb-SPIRE-KW is a weak nucleator." The authors should clearly indicate that this has been performed using rabbit skeletal actin, and not organism-specific actin proteins. Still, I agree that the number of WH2 domains is already strongly indicating whether a SPIRE protein will be a 'good' or 'weaker' actin nucleator.

- Related to the previous point: how identical are the actins isoforms in between these organisms. It would be interesting to discuss and document this, to better discuss the importance of SPIRE on actin dynamics.

In the revised manuscript we have addressed the differences in the WH2 domain numbers of choanoflagellate and mammalian SPIRE proteins experimentally by generating a mutant mouse SPIRE1 protein which has only two WH2 domains (Mm-SPIRE1-KW \square B,C)(Fig. 2e, f). As suggested by the reviewer we also have taken into account that the difference in nucleation activity might be due to the fact that we have employed rabbit actin in our assays. To better address this point we have included a multiple protein sequence alignment for mouse (skeletal muscle and cytoplasmic), rabbit skeletal muscle and choanoflagellate actin and highlighted contact sites for WH2 domains and their conservation. Furthermore we did a protein structure prediction of *Monosiga brevicollis* actin followed by a structure alignment with rabbit skeletal muscle actin bound to a WH2 domain to visualize actin:WH2 contact sites (Supplementary Figure 4a,b).

- Actin disassembly assay: this could also be performed in TIRF microscopy to infer the molecular mechanism of accelerated disassembly (filament fragmentation, faster barbed or pointed end depolymerization) ?

Excellent point. As suggested by the referee, we have monitored filament disassembly by TIRF imaging. To our surprise, we did not observe any shrinkage of filaments in the presence of Mm-SPIRE1-KW and Mm-SPIRE1-KW \square B,C, as the reaction had apparently already taken place in the relatively short time span between preparing the protein mixture, filling it into the TIRF chamber, mounting the chamber on the stage and starting the movie (about 40 seconds in total). Thus, it was impossible for us to determine whether the filaments are preferentially depolymerizing from a given end. However, as now shown in Supplementary Fig. 3c, increasing concentration of Mm-SPIRE1-KW clearly coincided with shorter filaments, while Mm-SPIRE1-KW \square B,C, containing only two WH2 domains, was virtually not effective in this assay. We have therefore rephrased the result section on page 8 to:

"Finally, we examined the disassembly of preformed actin filaments in the presence of Mm-SPIRE1-KW and Mm-SPIRE1-KW \square B,C. Increasing the concentration of Mm-SPIRE1-KW from 20 to 500 nM promoted rapid filament disassembly, with hardly any filaments remaining at the highest concentration as opposed to Mm-SPIRE1-KW \square B,C, which lacks two of the four WH2 domains (Supplementary Fig. 3c).

- figures: please show individual experimental repeats as data points, not only the average and SEM values in graphs.

We included individual data points in the bar diagrams representing the quantification of TIRF microscopy studies (Figure 3b-e).

- I do not understand the conclusion "The protein complex may have contributed to the evolution of animals and their sophisticated networks of polarised and communicating cells by providing an extended toolkit of actin and myosin functions in intracellular transport processes." Since this machinery is also present in other holozoans, even though the multiplicity of SPIRE and RAB orthologs would expand the toolkit size. Can the authors explain more clearly how they would see the SPIRE/FMN/MYO as a driver of evolution limited to animals only?

We agree with the referee that this statement needs a deeper discussion. We therefore have included a potential contribution of a SPIRE/MYO5 transport mechanism to the evolution of animal multicellularity into the discussion section (pages 15, 16). We argue that the complex life cycle of choanoflagellates, which includes facultative multicellularity, might indicate that a basic cell biological machinery for animal multicellularity has already been established in the single-celled ancestors of the animals. An interesting future experiment to evaluate the role of SPIRE in this will be to test if a SPIRE/MYO5 transport mechanism contributes to facultative multicellularity of choanoflagellates.

Minor point:

There are a number of writing issues along the text, like (p7) "Actin nucleation was found to be required an array of at least two WH2 domains, ...", (p8) "Mouse SNAP-SPIRE1- KW on the other hand induced actin filament assembly much more efficient already at lower concentration, ...", (p15) "which serve as tracks for myosin motor protein mediated centrifugal dispersion of the vesicles ..."

We are very sorry for the writing issues. The revised manuscript has been edited extensively and specifically also in respect to the writing.

Reviewer #2 (Remarks to the Author):

In this manuscript, Kollmar et al. investigate the evolutionary origins of SPIRE actin nucleators. They uncover SPIRE homologs in two unicellular organisms, choanoflagellates and ichthyosporeans. The choanoflagellate SPIRE is conserved enough to rescue the function of mammalian SPIRE in a knockdown system. Choanoflagellate SPIRE nucleates actin polymerization and maintains interaction partners essential for SPIRE function in mammals, including formin and class-5 myosin. Additionally, SPIRE co-localizes with RAB8, suggesting it forms a protein complex similar to that seen with SPIRE1 and RAB27A in mammals. Together, these findings show the SPIRE/myosin/formin protein complex originated in unicellular ancestors and has a conserved function in exocytic membrane transport.

This work provides important insight into the evolutionary origins of a core member of membrane trafficking by taking an interdisciplinary approach. The authors leveraged multiple in vivo and in vitro systems to demonstrate that SPIRE function is more ancient than previously appreciated. These findings will interest both those who study membrane transport and evolutionary biologists. However, the paper is not written for a broad readership and could significantly improve in clarity and organization. Additionally, some data detract from the core findings. Our specific comments are as follows:

Major comments:

1. We found that the organization of the manuscript, particularly the results sections pertaining to Figures 3 and 4, needs to be easier to follow, especially for someone outside the field. It seems the data in Figures 3 and 4 could be better organized by separating the findings for each actin regulator by figure (rather than describing all interactions in one figure). Details also appeared to be repetitive, appearing in the introduction and the results. Given the length of the intro, which seems more like a review, information could be introduced where it is most pertinent in the results sections.

This is a helpful comment. To get a better overview on different protein interactions we re-organized the data previously shown in Figures 3 and 4 and now prepared one Figure each for analyzing SPIRE/FMN interactions (Figure 4) and SPIRE/RAB interactions (Figure 5), and divided data on SPIRE and MYO5 into two Figures (Figure 6 and 7).

We have extensively edited the writing of the manuscript, this also includes some of the repetitive information. As the manuscript combines evolutionary biology and molecular cell biology, we did our best to introduce the different topics in a way that readers of both fields will profit. What in some cases seems to be repetitive, should help the non-expert to keep track.

2. In the evolutionary analysis, it should be made clear in the text that the protein sequences are classified as SPIRE1-3 based on percent identity and phylogenetic analysis. It was not shown that the genes obtained by tBLASTn are syntenic; therefore, it is unknown if all 'SPIRE1' proteins (for example) are truly orthologous. Furthermore, domains are often described as 'SPIRE-like' or distinguished by color. However, it would be helpful to know the protein identity relative to whichever species the authors used as their reference.

The classification as SPIRE1-3 was done based on the phylogenetic analysis, not on percent identity. It could have been done already just by visual comparison of the sequences in the alignment (see screen shots below). The regions around the GTBM are very different (not shown), but even in the highly conserved regions, many sequence positions clearly distinguish the three SPIRE classes. We do not want to discuss all the different reasons for broken synteny here. However, the separation of the three classes dates back to about 500 million years ago. Considering the hypothetical case, that a SPIRE2 would be a SPIRE1 and vice versa in one of the species studied, the two SPIRE in this species would have to have evolved exactly the identical characteristic mutations, that all the other species evolved independently for their exactly opposite SPIRE 1 and SPIRE2. This is highly unlikely. Even if the synteny around the SPIRE1-3 is not conserved in all species, they are functional orthologous based on their sequence characteristics.

There is no "SPIRE domain". SPIRE are multi-domain proteins and none of the multiple domains is termed SPIRE.

We termed those sequences "SPIRE-like", that share many but not all of the domains that the first identified SPIRE sequences (from human and fruitfly) consist of. For example, the amoebae SPIRE-like proteins do not contain the GTBM and FYVE domains but instead have several armadillo repeats at their N-terminal extension. Although SPIRE1-3 have the same domain organization, they have distinguished functions. This demonstrates the old problem of how to name and classify multi-domain proteins. Myosin and kinesins, for example, are classified and named based on just the presence of a single domain, the respective motor domain. Still, a class of myosins containing chitin synthase domains at their C-termini and being present only in fungi, are classified as chitin synthases (with the oddity of having N-terminal myosin motor domains) in the community working with fungi, although being referred to as myosin class-17 in the motor protein community. We think, that terming those sequences not sharing all SPIRE domains "SPIRE-like" will be the best way to later allow re-naming if experiments show that these SPIRE-like do not share any functions with the SPIRE.

3. In Figure 5, the authors report GFP-tagged SPIREs rescue melanosome dispersion. However, following siRNA knockdown, cells with melanosome dispersion do not always appear GFP fluorescent. For example, the image of Mb-SPIRE melanosomes shows a cell in the upper left corner with hyperdispersion, yet it does not appear to express GFP-tagged Mb-SPIRE.

We have revised figure 5 to present fluorescence signal using the 'Fire', rather than grayscale, look up table. We hope that the reviewer agrees that this more clearly shows cells in which protein expression is low (such as the one they mention above) as well as the range of expression between and within GFP-SPIRE expressing cells compared with grayscale.

The authors need to provide data, such as a western blot, to show how successful the knockdown was,

We have been unable to test protein expression as available antibodies for the detection of SPIRE1/2 by Western blot were not suitable to detect SPIRE1/2 at endogenous levels in melanocytes. However, using qPCR we have previously shown that knockdown using the oligonucleotides used in this study reduces expression of SPIRE1 and SPIRE2 to 22.0% and 18.7% of levels seen in control transfection using non-targeted oligonucleotides <https://pubmed.ncbi.nlm.nih.gov/32661310/> (Supplementary Figure 1a-c). We have included this information in a revised text for this section.

which may clarify the discrepancy in the image.

In these experiments (and previously <https://pubmed.ncbi.nlm.nih.gov/32661310/> (Supplementary Figure 1g) we only observed hyper-dispersion of melanosomes in cells expressing relatively low levels (close to the limit of detection) levels of SPIRE2 and Mb-SPIRE but not SPIRE1 or GFP. We hope that these revisions resolve this apparent discrepancy.

4. The analysis of human SPIREs and their Rab specificity detracted from the main focus on unicellular SPIRE. The fact that Choano SPIRE failed to interact with human Rab3 and Rab27, which evolutionarily arose after Rab8, was not surprising. Figure 6 could contribute to a different paper; alternatively, the framing of why it belongs in this story needs to be clarified.

We agree with the referee. We have now removed data related to differential SPIRE/RAB interactions shown in Figure 6 and will use them rather as a basis for a future manuscript. Mammalian RAB/SPIRE interaction data were contributed by Felix Straub, who is no longer an author of the manuscript.

Minor comments:

1. The authors note that the observed differences in Figure 2's in vitro work may be due to the use of rabbit actin. The authors should list the percent identities for Mb actin, mouse actin, and rabbit actin for readers to assess how likely the in vitro differences are due to species-specific actin differences.

Good point. We now state in the result section on page 8 "Our findings therefore strongly suggest that these differences are primarily caused by the difference in the number of WH2 domains, although we cannot formally exclude the possibility of differential binding of the choanoflagellate and mouse proteins to rabbit skeletal muscle actin used in our studies, despite the high similarity of choanoflagellate actin to rabbit actin (Supplementary Fig. 4b)".

We included a multiple protein sequence alignment for mouse (skeletal muscle and cytoplasmic), rabbit skeletal muscle and choanoflagellate actin and highlighted contact sites for WH2 domains and their conservation. Furthermore we did a protein structure prediction of *Monosiga brevicollis*

actin followed by a structure alignment with rabbit skeletal muscle actin bound to a WH2 domain to visualize actin:WH2 contact sites (Supplementary Figure 4a,b).

2. On page 12, the description of Myo5 motility in choanos seems to be an aside and is not pertinent to the story of SPIRE.

We agree and have now removed the data and description of Myo5 motility from the manuscript.

3. Please define acronyms when used the first time (for example, 'WH2' on page 3).

We addressed this point throughout the manuscript.

4. Some comments made throughout the results section are better placed in the discussion.

In the revision of the manuscript writing we have considered this comment and we hope that the manuscript has improved in this respect.

5. The schematic for Amoebozoa in Figure 1B and C should match for clarity.

We changed the schematic in Figure 1b accordingly.

6. Figure 1: It may be better to note the WH2 domains as A-D in the text, rather than by different shades of blue.

We have marked the WH2 domains in Fig. 1a with A-D in addition to the coloring.

7. In Figure 3A, why is mouse Formin and SPIRE used, yet Human Myo5a is shown?

To stay consistent with species we now show only human isoforms as vertebrate representatives for SPIRE, FMN and MYO5 proteins in comparison to *Monosiga brevicollis* proteins.

8. The model in Figure 3F was never referenced in the text, yet it was useful. Its description may help when clarifying the story's findings.

We are sorry that we did not reference the model figure in the text and have corrected this. We agree with the referee that the model figure is important and helps to understand SPIRE function in actomyosin transport processes.

9. The model in Figure 4D needs better labeling, such as the 'apical end.'

We worked on the schematic labelling and added labels for "apical pole", "basal pole", "flagellum", "collar" and "filopodia".

Reviewer #3 (Remarks to the Author):

Review to

“Actomyosin organelle functions of SPIRE actin nucleators precede animal evolution”

Cytoskeleton-based cargo transport exists in widely diverse organisms. Understanding the evolution of the proteins involved can improve the understanding of the transport process. Although the role of SPIRE protein in coordinating actin filament assembly and myosin motor-dependent force generation is well-studied in specific metazoans (mammals, flies), the evolutionary origin of the SPIRE remains unclear. Here, Kollmar et al. studied the origin of SPIRE by phylogenetic and biochemical approaches, revealing that SPIRE function originated in the last common ancestor of holozoans, which includes animals and their closest single-celled relatives, choanoflagellates. The data shown support that SPIRE from choanoflagellates is able to interact with formin, MYO5, and Rab GTPase, which also shares the similar function as mouse SPIRE protein. Overall, the data shown by the authors support their conclusions and clarifies the origin and function of the SPIRE protein.

This manuscript provides an interesting analysis of SPIRE across a wide evolutionary range, and as such is a very useful contribution to our knowledge of these proteins. There are several issues that require resolution, however.

Comments:

1. The authors mentioned that SPIRE is not universally conserved among Holozoa, e.g., quote ‘SPIRE has been lost in the unicellular filasterean’. The authors may want to clarify for this point: is the SPIRE sequence clearly not present in the filasterean genome or it is not accessible in the database?

As mentioned in the Results and Methods section, we did not rely on databases of annotated genes, but searched genomes and transcriptomes directly. Therefore, each statement refers to "not present in the genome". We think, and hope this reviewer agrees, that it is not necessary to repeat this for each individual statement on the phylogenetic distribution.

2. The illustration in Fig. 1b is confusing for the SPIRE part, since there are differences between SPIRE isoforms, especially for the WH2 domain, I would not recommend drawing exactly two WH2 diamonds there. Instead, the author may use a box to indicate all WH2s to avoid confusion.

We agree with the referee and now show a single box in each schematic representing the complete WH2-cluster instead of showing individual WH2 domains.

3. Fig. 2 c,d. These actin depolymerization assays need more controls:

a. Phalloidin: as a depolymerization-blocking control

b. LatA or LatB: This would show the effect of a compound that only sequesters.

As suggested, we have used additional controls. Since depolymerization occurs mainly at the barbed end, we used heterodimeric capping protein, but not phalloidin as it stabilizes the entire filament. LatB was used as a sequestering agent. These new results have been added to the manuscript and can be found in Fig. 2.

4. In Fig. 3d (pull-down assay), there are two issues:

- the pull-down assay shows a relatively weak binding, while the authors state the FMN-FSI protein strongly interacted with Mb-SPIRE. More conservative wording would be better
- one reason for weaker binding could be due to Mb-SPIRE autoinhibition by interacting KIND domain with FYVE2 domain. The authors may want to pulldown Mb-FMN and RAB8 with SPIRE-KW or SPIRE-FYVE2.

We agree with the referee and used more conservative wording for the interactions.

We repeated our pulldown studies using N-terminal Mb-SPIRE proteins (KIND-WH2) for Mb-FMN interactions (Figure 4e) and C-terminal Mb-SPIRE proteins (without KIND and WH2) for Mb-RAB8 interactions (Figure 5f).

A detailed analysis of a potential backfolding and auto-inhibition mechanism of the Mb-SPIRE protein, in analogy to the vertebrate SPIRE proteins, was, in our view, beyond the scope of this paper. However, considering the referee's comment, we did some preliminary studies to unravel such mechanisms which are presented below. Indeed, we found at least a weak interaction of GST-Mb-SPIRE-KIND and C-terminal GFP-Mb-SPIRE-ΔKW (panel b), which however needs to be further investigated.

5. For Fig. 3e colocalization assay, the author should use SPIRE-ΔKW and FMN-FH2-FSI as a good negative control.

Good point. We now expanded all colocalization studies shown before and included all control conditions that have been missed (Figures 4f, 5g, 6g). Those include SPIRE-ΔKW proteins as a FMN colocalization control, as well as co-expressions with fluorescent proteins only.

6. It is not clear how the data of Fig. 4c alone supports the conclusion that the mammalian and Choanoflagellate GTBMs are distantly related. More explanation would be required.

We wrote in the Results section: "Comparison of the choanoflagellate Mb/Sr-SPIRE MYO5 interaction motifs with vertebrate SPIRE GTBMs shows only low sequence homology (Fig. 4c).

However, the array of basic (blue), hydrophobic (green) and acidic residues (red) in the vertebrate GTBMs appears to be conserved in the Mb-SPIRE GTBM (Fig. 4c)." The GTBM is not a structured region in the sense that it folds into secondary structure elements and that these elements are part of a tertiary structure or the folding of the rest of the SPIRE. The GTBM binds to the surface of the myosin-GTD domain, and such interactions are not characterised by highly conserved residues, but by the conservation of the physicochemical properties of the respective residue segment. This is the case here. We show the functional similarity of the Mb-GTBM with the human SPIRE-GTBM. Thus, there is a functional correlation, but sequence homology only at the level of the set of physicochemical properties.

7. For the rescue experiment Fig. 5a, there are two concerns:

a. The Melanosomes in WT melanocytes must also be shown. Showing melanosomes in the rescue group is not sufficient;

To address this, we have revised Figure 5a by adding a phase contrast image of wild-type non-transfected murine melanocytes.

b. It is interesting that in Hs-SPIRE2 rescue group, on the left hand, there is one cell doesn't show any SPIRE/GTP signal, but shows a hyper-dispersed phenotype. Can the authors explain why?

As mentioned in our response to point 3 raised by reviewer 2 we have revised figure 5 to present fluorescence signal using the 'Fire', rather than grayscale, look up table. This more clearly shows cells in which protein expression is low as well as the range of expression between and within GFP-SPIRE expressing cells compared with grayscale. This confirms that the cell highlighted by this reviewer expresses low levels of SPIRE protein.

8. Since the authors compared the binding of SPIREs with formin, can the authors briefly describe the difference between SPIRE1-3 in the introduction and analyze which SPIRE isoform is more close to the MbSPIRE?

All three SPIRE (e.g. 1-3) have exactly the same phylogenetic distance to MbSPIRE. They are the result of two gene duplications at the origin of vertebrates, as described in the manuscript. The duplication occurred hundreds of millions of years after the separation of choanoflagellates from the Metazoa. Thus, at the sequence level, none of the SPIRE isoforms is closer to MbSPIRE than the others. In general, after a gene duplication, the functions (or cellular distribution and developmental expression) of the original single gene are shared between the two new genes. Therefore, on a functional level, none of the duplicated genes is closer to the original gene than the other. A closer relationship between one of the isoforms and the original gene could only exist if one of the isoforms diverges very widely, e.g. by gaining additional domains and losing existing domains. In this case, however, the functions of the previous gene are not split, but are retained in one of the genes, while the other develops new functions. This is not the case here.

REVIEWERS' COMMENTS:

Reviewer #1 (Remarks to the Author):

I am satisfied with the answers made by the authors for the points raised by the 3 reviewers. I think the manuscript has improved substantially, even though a slightly shorter and condensed version could have been appreciated.

Reviewer #2 (Remarks to the Author):

The revision is a significant improvement from the initial version. The new in vitro data and structural insights throughout the figures are nice additions, and the new layout of figures 4-6 has made the manuscript much clearer.

My only comment concerns point # 3. The 'Fire' lookup table helps, yet please add the word "low" to the following sentence on page 14: "Interestingly, we observed that in a subset of cells, [low] expression of Mb-SPIRE caused a hyper-dispersion phenotype, similar to that seen in experiments with SPIRE2." Otherwise, it is confusing that the most highly SPIRE-expressing cells show no hyperdispersion.

Overall, I am satisfied with the revisions and think it will be a noteworthy contribution to the field!

Reviewer #3 (Remarks to the Author):

The authors have addressed my concerns.